# Variational Gaussian Processes with Decoupled Conditionals

Xinran Zhu[1]    Kaiwen Wu[2]    Natalie Maus[2]    Jacob R. Gardner[2]    David Bindel[1]

[1]Cornell University    [2]University of Pennsylvania

{xz584,bindel}@cornell.edu {kaiwenwu,nmaus,jacobrg}@seas.upenn.edu

## Abstract

Variational Gaussian processes (GPs) approximate exact GP inference by using a small set of inducing points to form a sparse approximation of the true posterior, with the fidelity of the model increasing with additional inducing points. Although the approximation error in principle can be reduced by using more inducing points, this leads to scaling optimization challenges and computational complexity. To achieve scalability, inducing point methods typically introduce conditional independencies and then approximations to the training and test conditional distributions. In this paper, we consider an *alternative* approach to modifying the training and test conditionals, in which we make them more flexible. In particular, we investigate decoupling the parametric form of the predictive mean and covariance in the conditionals, and learn independent parameters for predictive mean and covariance. We derive new evidence lower bounds (ELBO) under these more flexible conditionals, and provide two concrete examples of applying the decoupled conditionals. Empirically, we find this additional flexibility leads to improved model performance on a variety of regression tasks and Bayesian optimization (BO) applications.

## 1 Introduction

Gaussian processes (GPs) are a powerful class of non-parametric probabilistic models [38]. Their flexibility and ability to make probabilistic predictions make them particularly useful in situations where uncertainty estimates are important [23, 30, 43, 45, 56]. Although elegant, exact GPs become intractable for large datasets since the computational cost scales cubically with the size of the dataset.

To overcome this limitation, various sparse approximate GP approaches have been developed, mostly relying on sparse approximations of the true posterior [28, 29, 36]. Among those, variational GPs have become increasingly popular because they enable stochastic minibatching and apply to both regression [17] and classification [18] tasks. However, the approximate posterior and variational inference leave an unavoidable gap from the exact model. Especially with too few inducing points, variational GPs could yield suboptimal accuracy performance due to a lack of expressiveness. Using more inducing points closes this gap in principle, but leads to additional computational complexity and optimization difficulties [15]. While several works have studied ways of allowing more or better placed inducing points to improve accuracy of variational GPs, [34, 41, 50, 57], it is nevertheless challenging to use more than a small fraction of the dataset size in inducing points.

In this work, we take a different approach to improving model expressiveness for better accuracy of variational GPs through *decoupling conditionals*. To start, we point out that inducing point approximations rely on two key conditionals (see Sec. 3.1) – the training and the test conditional, where the mean and covariance are parameterized by inducing points and kernel hyperparameters. Typically, prior works have focused on various *relaxations* of these conditionals that enable significant computational complexity benefits [36, 44, 46, 47, 48]. In this work, we consider alternative changes

37th Conference on Neural Information Processing Systems (NeurIPS 2023).

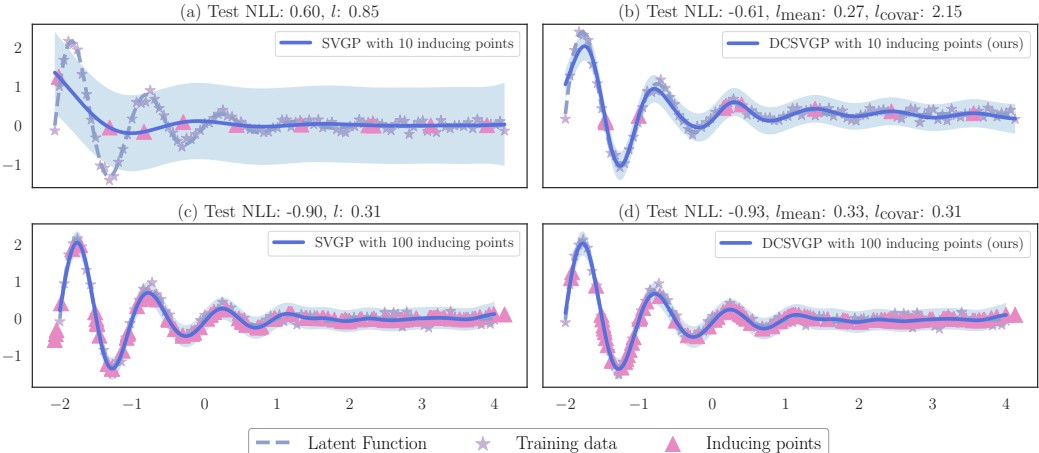

Figure 1: We compare model fit on a 1D latent function using 100 training samples. Solid curves with shading area depict the predictive mean and 95% confidence interval. Using 10 inducing points, in **subplot (a)** SVGP underfits the latent function with large lengthscale $l = 0.85$; while in **subplot (b)**, our DCSVGP model (see Sec. 3.2) fits better and learns different decoupled lengthscales $l_{\text{mean}}$ and $l_{\text{covar}}$. Using 100 inducing points, in **subplot (c) and (d)**, both models fits well with similar lengthscales around 0.3. See Sec. 1 for more details.

that increase the flexibility of these conditionals. In Sec. 3.2, we describe two concrete examples of this idea: decoupling kernel lengthscales and decoupling entire deep feature extractors.

As a simple illustration, Fig. 1 illustrates, for example, how decoupled lengthscales improve model fitting. Our model DCSVGP (see Sec. 3.2) learns decoupled lengthscales $l_{\text{mean}}$ and $l_{\text{covar}}$ for mean and covariance respectively, and we compare with baseline SVGP (see Sec. 2) which learns one lengthscale $l$.

To summarize our contributions: 1) We propose decoupled conditionals in variational GPs to improve model expressiveness for better accuracy. 2) We show that our idea is compatible with the variational framework and rigorously derive an ELBO for variational inference with decoupled conditionals. 3) We provide two concrete examples of applying decoupled conditionals and empirically show the superior performance of our models through extensive regression tasks and BO applications.

## 2  Background

We assume familiarity with GPs [38] and briefly introduce them for notational clarity. Given observation locations $\mathbf{X} = \{\mathbf{x}_i\}_{i=1}^n \subset \mathbb{R}^d$, a GP prior induces a multivariate Normal belief for latent function values $\mathbf{f} = \{f(\mathbf{x}_i)\}$: $\mathbf{f} \sim \mathcal{N}(\mu_{\mathbf{X}}, \mathbf{K}_{nn})$, where $\mu_{\mathbf{X}}, \mathbf{K}_{nn}$ are the mean values and covariance matrix at data $\mathbf{X}$. Givend observations $\mathbf{y} = \mathbf{f} + \epsilon$ with Gaussian noise $\epsilon \sim \mathcal{N}(0, \sigma^2 I)$, the posterior distribution of the function value $\mathbf{f}^*$ at a new data point $x^*$ is $p(\mathbf{f}^*|\mathbf{y}) = \mathcal{N}(\mu^*, \Sigma^{**})$, where

$$\mu^* = \mu(\mathbf{x}^*) + \mathbf{K}_{*n}(\mathbf{K}_{nn} + \sigma^2 \mathbf{I})^{-1}(\mathbf{y} - \mu_{\mathbf{X}}),$$
$$\Sigma^{**} = \mathbf{K}(\mathbf{x}^*, \mathbf{x}^*) - \mathbf{K}_{*n}(\mathbf{K}_{nn} + \sigma^2 \mathbf{I})^{-1}\mathbf{K}_{*n}^T.$$

Model hyperparameters such as kernel hyperparameters and noise $\sigma$ are typically estimated by Maximum Likelihood using standard numerical solvers such as LBFGS [35]. If no approximations are used, each evaluation to optimize the log marginal likelihood function costs $\mathcal{O}(n^3)$ flops and $\mathcal{O}(n^2)$ memory, thus motivating approximate methods for large training datasets.

### 2.1  Sparse Gaussian Processes

To overcome the scalability limitations of exact GPs, many authors have proposed a variety of sparse GPs by introducing *inducing points* $\mathbf{Z} = \{\mathbf{z}_i\}_{i=1}^m$ [17, 44, 47, 48, 49]. Inducing points are associated with inducing values $\mathbf{f}_m$, which represent latent function values at $\mathbf{Z}$ under the same GP assumption.

Although inducing values $\mathbf{f}_m$ are marginalized out in the predictive distribution, they typically reduces training costs from $\mathcal{O}(n^3)$ to $\mathcal{O}(n^2 m + m^3)$ for each gradient step, where $m \ll n$.

**SGPR.** Based on inducing point methods, Sparse Gaussian Process Regression (SGPR) [49] further introduces variational inference into sparse GPs. Assuming that inducing data is adequate for inference, Titsias [49] further introduces a variational distribution $\phi(\mathbf{f}_m) \sim \mathcal{N}(\mathbf{m}, \mathbf{S})$ that approximates the posterior $p(\mathbf{f}_m|\mathbf{y})$. Thus, the predictive density can be approximated as

$$q(\mathbf{f}^*) = \int p(\mathbf{f}^*|\mathbf{f}_m)p(\mathbf{f}_m|\mathbf{y})d\mathbf{f}_m = \int p(\mathbf{f}^*|\mathbf{f}_m)\phi(\mathbf{f}_m)d\mathbf{f}_m = \mathcal{N}(\mu_{\mathbf{f}}(\mathbf{x}^*), \sigma_{\mathbf{f}}(\mathbf{x}^*)^2).$$

Here the predictive mean $\mu_{\mathbf{f}}(\mathbf{x}^*)$ and the latent function variance $\sigma_{\mathbf{f}}(\mathbf{x}^*)^2$ are:

$$\mu_{\mathbf{f}}(\mathbf{x}^*) = \mathbf{K}_{*m}\mathbf{K}_{mm}^{-1}\mathbf{m}, \ \ \sigma_{\mathbf{f}}(\mathbf{x}^*)^2 = \tilde{\mathbf{K}}_{**} + \mathbf{K}_{*m}\mathbf{K}_{mm}^{-1}\mathbf{S}\mathbf{K}_{mm}^{-1}\mathbf{K}_{m*}, \tag{1}$$

where $\tilde{\mathbf{K}}_{**} = \mathbf{K}_{**} - \mathbf{K}_{*m}\mathbf{K}_{mm}^{-1}\mathbf{K}_{*m}^T$. The variational distribution $\phi(\mathbf{f}_m) = \mathcal{N}(\mathbf{m}, \mathbf{S})$ is then learned by maximizing the variational ELBO [18, 21], which is a lower bound on the log marginal likelihood:

$$\log p(\mathbf{y}) \geq \mathbb{E}_{q(\mathbf{f})}[\log p(\mathbf{y}|\mathbf{f})] - \mathrm{KL}[\phi(\mathbf{f}_m)||p(\mathbf{f}_m)],$$

where $q(\mathbf{f}) = \int p(\mathbf{f}|\mathbf{f}_m)\phi(\mathbf{f}_m)d\mathbf{f}_m$, and $\mathrm{KL}[\phi(\mathbf{f}_m)||p(\mathbf{f}_m)]$ is the KL divergence [27].

**SVGP.** To enable data subsampling during training, SVGP [17] avoids analytic solutions to the ELBO, but rather decomposes the ELBO as a sum of losses over training labels and enables stochastic gradient descent (SGD) [39] training:

$$\mathrm{ELBO}_{\mathrm{SVGP}} = \sum_{i=1}^{n} \left\{ \log \mathcal{N}(y_i|\mu_{\mathbf{f}}(\mathbf{x}_i), \sigma^2) - \frac{\sigma_{\mathbf{f}}(\mathbf{x}_i)^2}{2\sigma^2} \right\} - \beta\mathrm{KL}\left[\phi(\mathbf{f}_m)||p(\mathbf{f}_m)\right], \tag{2}$$

where $\mu_{\mathbf{f}}(\cdot), \sigma_{\mathbf{f}}(\cdot)^2$ are the predictive mean and latent function variance from Eq. 1 respectively, and $\beta$ is an optional regularization parameter for the KL divergence and can be tuned in practice [22].

**PPGPR.** To achieve heteroscedastic modeling and improve predictive variances, the Parametric Gaussian Process Regressor (PPGPR) [22] targets the preditive distribution directly in the loss function. It shares the same predictive mean $\mu_{\mathbf{f}}(\cdot)$ and latent function variance $\sigma_{\mathbf{f}}(\cdot)^2$ as SVGP, but uses a slightly different stochastic ELBO loss:

$$\mathrm{ELBO}_{\mathrm{PPGPR}} = \sum_{i=1}^{n} \log \mathcal{N}(y_i|\mu_{\mathbf{f}}(\mathbf{x}_i), \sigma^2 + \sigma_{\mathbf{f}}(\mathbf{x}_i)^2) - \beta\mathrm{KL}[\phi(\mathbf{f}_m)||p(\mathbf{f}_m)].$$

**Whitening.** In practice, the variational distribution $\phi(\mathbf{f}_m)$ is usually "whitened" to accelerate the optimization of the variational distribution [32]. Conventionally, the whitened variational distribution is $\tilde{\phi}(\mathbf{f}_m) = \mathcal{N}(\tilde{\mathbf{m}}, \tilde{\mathbf{S}}) = \mathcal{N}(\mathbf{K}_{mm}^{-1/2}\mathbf{m}, \mathbf{K}_{mm}^{-1/2}\mathbf{S}\mathbf{K}_{mm}^{-1/2})$, where $\mathbf{K}_{mm}^{1/2}$ is the square root of $\mathbf{K}_{mm}$. With whitening, the KL divergence and predictive distribution in ELBO (Eq. 2) are both simplified:

$$\mathrm{KL}[\phi(\mathbf{f}_m)||p(\mathbf{f}_m)] = \mathrm{KL}(\tilde{\phi}(\mathbf{f}_m)||p_0(\mathbf{f}_m)), \text{ where } p_0(\mathbf{f}_m) = \mathcal{N}(\mathbf{0}, \mathbf{I}),$$

$$\mu_{\mathbf{f}}(\mathbf{x}_i) = \mathbf{K}_{im}\mathbf{K}_{mm}^{-1/2}\tilde{\mathbf{m}}, \ \ \sigma_{\mathbf{f}}(\mathbf{x}_i)^2 = \tilde{\mathbf{K}}_{ii} + \mathbf{K}_{im}\mathbf{K}_{mm}^{-1/2}\tilde{\mathbf{S}}\mathbf{K}_{mm}^{-1/2}\mathbf{K}_{mi}.$$

## 3 Methodology

Here, we present variational GPs with decoupled conditionals. In Sec. 3.1 and 3.2, we introduce decoupled conditionals under a unifying framework for approximate GPs, followed by two concrete examples. In Sec. 3.3, we derive an ELBO to do variational inference with decoupled conditionals.

### 3.1 Conditionals of Approximate GPs

Quinonero-Candela and Rasmussen [36] introduced a unifying framework for sparse GPs. Through the framework, an approximate GP can be interpreted as "an exact inference with an approximated

Table 1: Exact and approximate conditionals. See Sec. 3.1 and Sec. 3.2 for details. Here $\tilde{\mathbf{K}}_{nn} = \mathbf{K}_{nn} - \mathbf{K}_{nm}\mathbf{K}_{mm}^{-1}\mathbf{K}_{mn}$ and $\tilde{\mathbf{K}}_{**} = \mathbf{K}_{**} - \mathbf{K}_{*m}\mathbf{K}_{mm}^{-1}\mathbf{K}_{m*}$.

|  | Training Conditionals | Test Conditionals |
|---|---|---|
| DTC | $q(\mathbf{f}|\mathbf{f}_m) = \mathcal{N}(\mathbf{K}_{nm}\mathbf{K}_{mm}^{-1}\mathbf{f}_m, 0)$ | $p(\mathbf{f}^*|\mathbf{f}_m) = \mathcal{N}(\mathbf{K}_{*m}\mathbf{K}_{mm}^{-1}\mathbf{f}_m, \tilde{\mathbf{K}}_{**})$ |
| FITC | $q(\mathbf{f}|\mathbf{f}_m) = \mathcal{N}(\mathbf{K}_{nm}\mathbf{K}_{mm}^{-1}\mathbf{f}_m, \mathrm{diag}[\tilde{\mathbf{K}}_{nn}])$ | $p(\mathbf{f}^*|\mathbf{f}_m) = \mathcal{N}(\mathbf{K}_{*m}\mathbf{K}_{mm}^{-1}\mathbf{f}_m, \tilde{\mathbf{K}}_{**})$ |
| SVGP | $p(\mathbf{f}|\mathbf{f}_m) = \mathcal{N}(\mathbf{K}_{nm}\mathbf{K}_{mm}^{-1}\mathbf{f}_m, \tilde{\mathbf{K}}_{nn})$ | $p(\mathbf{f}^*|\mathbf{f}_m) = \mathcal{N}(\mathbf{K}_{*m}\mathbf{K}_{mm}^{-1}\mathbf{f}_m, \tilde{\mathbf{K}}_{**})$ |
| Ours | $\psi(\mathbf{f}|\mathbf{f}_m) = \mathcal{N}(\mathbf{Q}_{nm}\mathbf{Q}_{mm}^{-1}\mathbf{f}_m, \tilde{\mathbf{K}}_{nn})$ | $\psi(\mathbf{f}^*|\mathbf{f}_m) = \mathcal{N}(\mathbf{Q}_{*m}\mathbf{Q}_{mm}^{-1}\mathbf{f}_m, \tilde{\mathbf{K}}_{**})$ |

prior" in contrast to "an approximate inference with an exact prior". With inducing points $\mathbf{Z} = \{\mathbf{z}_i\}_{i=1}^m$ and inducing values $\mathbf{f}_m$, most sparse GPs approximate the joint Gaussian prior as:

$$p(\mathbf{f}^*, \mathbf{f}) = \int p(\mathbf{f}^*, \mathbf{f}|\mathbf{f}_m)p(\mathbf{f}_m)d\mathbf{f}_m \approx \int q(\mathbf{f}^*|\mathbf{f}_m)q(\mathbf{f}|\mathbf{f}_m)p(\mathbf{f}_m)d\mathbf{f}_m = q(\mathbf{f}^*, \mathbf{f}),$$

where dependencies between training data $\mathbf{f}$ and test data $\mathbf{f}^*$ are *induced* by inducing values $\mathbf{f}_m$. Different approximations can be made of the inducing *training conditionals* $q(\mathbf{f}|\mathbf{f}_m)$ and the inducing *test conditionals* $q(\mathbf{f}^*|\mathbf{f}_m)$. Table 1 provides the exact expressions of the two conditionals and examples of approximations used in various approximate GP methods. For example, the Deterministic Training Conditional (DTC) approximation uses a deterministic training conditional and the exact test conditional [44]; The Fully Independent Training Conditional (FITC) approximation uses an approximate training conditional with diagonal corrections and the exact test conditional [48].

### 3.2 Decoupled Conditionals and Prediction

In this paper, we consider augmenting the exact conditionals by a more flexible form that decouples the kernel hyperparameters in the mean and covariance in both conditionals:

$$\begin{aligned} \text{training conditional } \psi(\mathbf{f}|\mathbf{f}_m) &= \mathcal{N}(\mathbf{Q}_{nm}\mathbf{Q}_{mm}^{-1}\mathbf{f}_m, \tilde{\mathbf{K}}_{nn}) \approx p(\mathbf{f}|\mathbf{f}_m), \\ \text{test conditional } \psi(\mathbf{f}^*|\mathbf{f}_m) &= \mathcal{N}(\mathbf{Q}_{*m}\mathbf{Q}_{mm}^{-1}\mathbf{f}_m, \tilde{\mathbf{K}}_{**}) \approx p(\mathbf{f}^*|\mathbf{f}_m), \end{aligned} \quad (3)$$

where the $\mathbf{Q}$ and $\mathbf{K}$ matrices are formed by the same family of kernel functions, but some kernel hyperparameters are *decoupled*. See Table 1 for a comparison with other approximations. Decoupled conditionals improve model flexibility without relying on more inducing points and it applies to various SVGP-based models. We provide two examples of *decoupled* models that we will evaluate.

**Example 1: Decoupled lengthscales.** By decoupling the kernel lengthscale $l$ into $l_{\mathrm{mean}}$ for the mean and $l_{\mathrm{covar}}$ for the covariance, we enable the model to learn in settings where the function value changes more rapidly than the variance. In Eq. 3, this corresponds to the $\mathbf{Q}$ and $\mathbf{K}$ kernel matrices being formed using separate lengthscales. For example, an RBF kernel gives

$$\mathbf{Q}_{mm} = \left[\exp\left(-\frac{\|\mathbf{z}_i - \mathbf{z}_j\|^2}{2l_{\mathrm{mean}}^2}\right)\right]_{\mathbf{z}_i, \mathbf{z}_j \in \mathbf{Z}}, \quad \mathbf{K}_{mm} = \left[\exp\left(-\frac{\|\mathbf{z}_i - \mathbf{z}_j\|^2}{2l_{\mathrm{covar}}^2}\right)\right]_{\mathbf{z}_i, \mathbf{z}_j \in \mathbf{Z}}.$$

We will denote the application of decoupled lengthscales to SVGP as **Decoupled Conditional SVGP (DCSVGP)**.

**Example 2: Decoupled deep kernel learning.** Wilson et al. [53] proposed deep kernel learning (DKL) which stacks a deep neural network feature extractor $h$ with a GP layer. Combining DKL with variational GP models is straightforward, with the feature extractor $h$ learned through ELBO with all other model (hyper)parameters [54]. The feature extractor $h$ can then be decoupled: one $h_{\mathrm{mean}}$ for the mean and one $h_{\mathrm{covar}}$ for the covariance. Again using the RBF kernel as a base example:

$$\mathbf{Q}_{mm} = \left[\exp\left(-\frac{\|h_{\mathrm{mean}}(\mathbf{z}_i) - h_{\mathrm{mean}}(\mathbf{z}_j)\|^2}{2l^2}\right)\right]_{\mathbf{z}_i, \mathbf{z}_j \in \mathbf{Z}},$$

$$\mathbf{K}_{mm} = \left[\exp\left(-\frac{\|h_{\mathrm{covar}}(\mathbf{z}_i) - h_{\mathrm{covar}}(\mathbf{z}_j)\|^2}{2l^2}\right)\right]_{\mathbf{z}_i, \mathbf{z}_j \in \mathbf{Z}}.$$

We denote the application of decoupled deep feature extractors to SVGP as **SVGP-DCDKL**.

**Prediction.** Using decoupled conditionals, the predictive posterior at a new point $\mathbf{x}^*$ is

$$q(\mathbf{y}^*) = \int p(\mathbf{y}^*|\mathbf{f}^*)\psi(\mathbf{f}^*|\mathbf{f}_m)\phi(\mathbf{f}_m)\,d\mathbf{f}_m d\mathbf{f}^* = \mathcal{N}(\mathbf{y}^*|\tilde{\mu}_\mathbf{f}(\mathbf{x}^*), \tilde{\sigma}_\mathbf{f}(\mathbf{x}^*)^2 + \sigma^2\mathbf{I}),$$

where the predictive mean $\tilde{\mu}_\mathbf{f}(\mathbf{x}^*)$ and latent function variance $\tilde{\sigma}_\mathbf{f}(\mathbf{x}^*)^2$ are similar to Eq. 1:

$$\tilde{\mu}_\mathbf{f}(\mathbf{x}^*) = \mathbf{Q}_{*m}\mathbf{Q}_{mm}^{-1}\mathbf{m}, \quad \tilde{\sigma}_\mathbf{f}(\mathbf{x}^*)^2 = \tilde{\mathbf{K}}_{**} + \mathbf{Q}_{*m}\mathbf{Q}_{mm}^{-1}\mathbf{S}\mathbf{Q}_{mm}^{-1}\mathbf{Q}_{m*}. \tag{4}$$

### 3.3 The Evidence Lower Bound (ELBO)

In this section, we derive an ELBO for DCSVGP model fitting (other examples of decoupled models follow the same derivation). All model parameters and hyperparameters are learned by maximizing the resulting ELBO. Following the standard variational inference [2], we approximate the true posterior distribution $p(\mathbf{f}, \mathbf{f}_m|\mathbf{y})$ by the variational distribution $q(\mathbf{f}, \mathbf{f}_m)$ and miminize the KL divergence: $\min \mathrm{KL}(q(\mathbf{f}, \mathbf{f}_m)||p(\mathbf{f}, \mathbf{f}_m|\mathbf{y}))$. In the standard case, $q(\mathbf{f}, \mathbf{f}_m) = p(\mathbf{f}|\mathbf{f}_m)\phi(\mathbf{f}_m)$, but a decoupled model has $q(\mathbf{f}, \mathbf{f}_m) = \psi(\mathbf{f}|\mathbf{f}_m)\phi(\mathbf{f}_m)$ since it further approximates the training conditional $p(\mathbf{f}|\mathbf{f}_m)$ by a decoupled one $\psi(\mathbf{f}|\mathbf{f}_m)$. This difference leads to the following ELBO for the decoupled model:

$$\log(p(\mathbf{y})) \geq \mathrm{ELBO}(q) = \mathbb{E}[\log(p(\mathbf{y}|\mathbf{f}, \mathbf{f}_m))] - \mathrm{KL}(q(\mathbf{f}, \mathbf{f}_m)||p(\mathbf{f}, \mathbf{f}_m))$$

$$= \sum_{i=1}^{n}\left\{\log\mathcal{N}(y_i|\tilde{\mu}_\mathbf{f}(\mathbf{x}_i), \sigma^2) - \frac{\tilde{\sigma}_\mathbf{f}(\mathbf{x}_i)^2}{2\sigma^2}\right\} - \int \psi(\mathbf{f}|\mathbf{f}_m)\phi(\mathbf{f}_m)\log\frac{\psi(\mathbf{f}|\mathbf{f}_m)\phi(\mathbf{f}_m)}{p(\mathbf{f}|\mathbf{f}_m)p(\mathbf{f}_m)}\,d\mathbf{f}d\mathbf{f}_m$$

$$= \sum_{i=1}^{n}\left\{\log\mathcal{N}(y_i|\tilde{\mu}_\mathbf{f}(\mathbf{x}_i), \sigma^2) - \frac{\tilde{\sigma}_\mathbf{f}(\mathbf{x}_i)^2}{2\sigma^2}\right\} - \mathrm{KL}(\phi(\mathbf{f}_m)||p(\mathbf{f}_m)) - \int \psi(\mathbf{f}|\mathbf{f}_m)\phi(\mathbf{f}_m)\log\frac{\psi(\mathbf{f}|\mathbf{f}_m)}{p(\mathbf{f}|\mathbf{f}_m)}\,d\mathbf{f}d\mathbf{f}_m$$

$$= \sum_{i=1}^{n}\left\{\log\mathcal{N}(y_i|\tilde{\mu}_\mathbf{f}(\mathbf{x}_i), \sigma^2) - \frac{\tilde{\sigma}_\mathbf{f}(\mathbf{x}_i)^2}{2\sigma^2}\right\} - \mathrm{KL}(\phi(\mathbf{f}_m)||p(\mathbf{f}_m)) - \underbrace{\mathbb{E}_{\phi(\mathbf{f}_m)}[\mathrm{KL}(\psi(\mathbf{f}|\mathbf{f}_m)||p(\mathbf{f}|\mathbf{f}_m))]}_{:=\Omega}.$$

We refer to App. A.1 for additional derivation details. Adding regularization parameters $\beta_1$ and $\beta_2$ to the KL divergence terms as is often done in practice, the ELBO for DCSVGP is

$$\mathrm{ELBO}_{\mathrm{DCSVGP}} = \sum_{i=1}^{n}\left\{\log\mathcal{N}(y_i|\tilde{\mu}_\mathbf{f}(\mathbf{x}_i), \sigma^2) - \frac{\tilde{\sigma}_\mathbf{f}(\mathbf{x}_i)^2}{2\sigma^2}\right\} - \beta_1\mathrm{KL}\left[\phi(\mathbf{f}_m)||p(\mathbf{f}_m)\right] - \beta_2\Omega \tag{5}$$

where the predictive mean $\tilde{\mu}_\mathbf{f}(\mathbf{x}_i)$ and latent function variance $\tilde{\sigma}_\mathbf{f}(\mathbf{x}_i)^2$ are same as Eq. 4

$$\tilde{\mu}_\mathbf{f}(\mathbf{x}_i) = \mathbf{Q}_{im}\mathbf{Q}_{mm}^{-1}\mathbf{m}, \quad \tilde{\sigma}_\mathbf{f}(\mathbf{x}_i)^2 = \tilde{\mathbf{K}}_{ii} + \mathbf{Q}_{im}\mathbf{Q}_{mm}^{-1}\mathbf{S}\mathbf{Q}_{mm}^{-1}\mathbf{Q}_{mi}. \tag{6}$$

**The explicit expression of $\Omega$.** The $\Omega$ term can be computed explicitly (see App. A.1):

$$\Omega = \mathbb{E}_{\phi(\mathbf{f}_m)}[\mathrm{KL}(\psi(\mathbf{f}|\mathbf{f}_m)||p(\mathbf{f}|\mathbf{f}_m))] = \frac{1}{2}\mathbb{E}[\mathbf{f}_m^T\mathbf{T}\mathbf{f}_m] = \frac{1}{2}\left(\mathrm{Tr}(\mathbf{T}\mathbf{S}) + \mathbf{m}^T\mathbf{T}\mathbf{m}\right),$$

where $\mathbf{T} = \mathbf{A}^T\tilde{\mathbf{K}}_{nn}^{-1}\mathbf{A}$, $\mathbf{A} = \mathbf{Q}_{nm}\mathbf{Q}_{mm}^{-1} - \mathbf{K}_{nm}\mathbf{K}_{mm}^{-1}$, and $\phi(\mathbf{f}_m) = \mathcal{N}(\mathbf{m}, \mathbf{S})$.

**Comparing $\mathrm{ELBO}_{\mathrm{DCSVGP}}$ and $\mathrm{ELBO}_{\mathrm{SVGP}}$.** The $\mathrm{ELBO}_{\mathrm{SVGP}}$ in Eq. 2 and $\mathrm{ELBO}_{\mathrm{DCSVGP}}$ in Eq. 5 both consist of an approximate likelihood term and a KL divergence part. There are two differences: 1) $\mathrm{ELBO}_{\mathrm{DCSVGP}}$ involves different predictive mean $\tilde{\mu}_\mathbf{f}(\mathbf{x}_i)$ and variance $\tilde{\sigma}_\mathbf{f}(\mathbf{x}_i)^2$ derived from decoupled conditionals, see Eq. 6; 2) $\mathrm{ELBO}_{\mathrm{DCSVGP}}$ contains an additional KL divergence term $\Omega$, which is an expected KL divergence of the two training conditionals over the variational distribution $\phi(\mathbf{f}_m)$, regularizing the difference between the decoupled conditional $\psi(\mathbf{f}|\mathbf{f}_m)$ and the exact one $p(\mathbf{f}|\mathbf{f}_m)$.

**The regularization parameter.** In Eq. 5, $\mathrm{ELBO}_{\mathrm{DCSVGP}}$ contains two KL divergence terms with regularization parameters $\beta_1$ and $\beta_2$, respectively. Varying $\beta_1$ controls the regularization on prior similarities, same as SVGP, while varying $\beta_2$ controls the regularization on the difference between the decoupled conditional and the exact one. In the limit where $\beta_2 \to +\infty$, decoupling is disallowed and DCSVGP degenerates to SVGP. See Sec. 5.2 for more discussion and empirical study.

**Whitening.** Decoupling the mean and covariance hyperparameters introduces a challenge where one can whiten with either $\mathbf{K}_{mm}^{1/2}$ or $\mathbf{Q}_{mm}^{1/2}$, with the drawback of only simplifying one term. $\mathbf{K}$-whitening simplifies the KL divergence but leaves the predictive distribution significantly "less linear", while $\mathbf{Q}$-whitening does the opposite. For example, the predictive distribution from the two whitening choices are:

$$\mathbf{Q}\text{-whitening} \quad \tilde{\mu}_{\mathbf{f}}(\mathbf{x}_i) = \mathbf{Q}_{im}\mathbf{Q}_{mm}^{-1/2}\bar{\mathbf{m}}, \quad \tilde{\sigma}_{\mathbf{f}}(\mathbf{x}_i)^2 = \tilde{\mathbf{K}}_{ii} + \mathbf{Q}_{im}\mathbf{Q}_{mm}^{-1/2}\bar{\mathbf{S}}\mathbf{Q}_{mm}^{-1/2}\mathbf{Q}_{mi},$$

$$\mathbf{K}\text{-whitening} \quad \tilde{\mu}_{\mathbf{f}}(\mathbf{x}_i) = \mathbf{Q}_{im}\mathbf{Q}_{mm}^{-1}\mathbf{K}_{mm}^{1/2}\bar{\mathbf{m}}, \quad \tilde{\sigma}_{\mathbf{f}}(\mathbf{x}_i)^2 = \tilde{\mathbf{K}}_{ii} + \mathbf{Q}_{im}\mathbf{Q}_{mm}^{-1}\mathbf{K}_{mm}^{1/2}\bar{\mathbf{S}}\mathbf{K}_{mm}^{1/2}\mathbf{Q}_{mm}^{-1}\mathbf{Q}_{mi}.$$

Empirically $\mathbf{Q}$-whitening performs better because a simplified predictive distribution is more favorable for model fitting. See App. A.2 for KL divergence terms and derivation details.

# 4 Related Work

Variational GPs have been successfully extended to various settings [1, 3, 4, 11, 16, 19, 20, 31, 51, 55, 54]. Among these vast enhancements, much attention has been devoted to the study of inducing points, the core part that leads to an expressiveness-and-complexity trade-off. Many works have studied better placement of inducing points, including selecting from training data [9, 24, 42, 44, 47, 7, 34] and from the input domain [40, 50], but typically inducing points are learned as model parameters [17, 48]. On the other hand, several schemes have been developed to reduce complexity and allow more inducing points, e.g. Fourier methods [19, 29] and methods that decouple inducing points used in variational mean and covariance [8, 15, 41]. However, more inducing points could be less informative under certain data characteristics [6, 16, 50] or result in suboptimal model fitting [15, 57]. Therefore, our work takes a different direction to improve variational GPs via more flexible mean and covariance modeling rather than replying on more inducing points. The work that is most closely related to ours is the ODSVGP model [41] that decouples the inducing points for mean and variance to allow more inducing points used in mean modeling. With similar decoupling idea, we are motivated by increasing model flexibility rather than reducing model complexity and we simply decouple the kernel hyperparameters with negligible additional costs.

# 5 Experiments

We evaluate the performance of decoupled models proposed in Sec. 3.2: DCSVGP (variational GPs using decoupled lengthscales) and SVGP-DCDKL (variational GPs with deep kernel learning using decoupled deep feature extractors). Because PPGPR [22] is orthogonal to our decoupling method, similarly we also evaluate DCPPGPR (decoupled lengthscales) and PPGPR-DCDKL (decoupled feature extractors). All experiments use an RBF kernel and a zero prior mean and are accelerated through GPyTorch [14] on a single GPU. Code is available at `https://github.com/xinranzhu/Variational-GP-Decoupled-Conditionals`.

## 5.1 Regression Tasks

We consider 10 UCI regression datasets [10] with up to 386508 training examples and up to 380 dimensions. We present main results with non-ARD RBF kernels, the $\mathbf{Q}$-whitening scheme described in Sec. 3.3, and set $\beta_2 = 1e\text{-}3$. Results are averaged over 10 random dataset splits. For additional results with ARD kernels and $\mathbf{K}$-whitening, see App. B.1.4 and B.1.5. For more experiment setup and training details such as the number of inducing points, we refer to App. B.1.

**DCSVGP and DCPPGPR.** We first compare DCSVGP to baseline SVGP [17] and ODSVGP [41] and similarly compare DCPPGPR to PPGPR [22] and ODPPGPR. Table 2 shows the test RMSE and NLL results. We observe that DCSVGP (and similarly DCPPGPR) yields the lowest RMSE and NLL on 8 datasets, demonstrating the improved flexibility of our method. We report all learned lengthscales in App. B.1.1. On the other hand, we also note that on Protein and Elevators, DCPPGPR slightly overfits with worse test NLL. We show in Sec. 5.2 that increasing $\beta_2$ resolves this issue. We also find that decoupling lengthscales generally improves model calibration (see App. B.1.3 for results and discussions).

Table 2: Test RMSE and NLL on 10 regression datasets (lower is better). Results are averaged over 10 random train/validation/test splits. Statistical significance is indicated by **bold**. See the supplement for standard errors.

| Test RMSE | Pol | Elevators | Bike | Kin40k | Protein | Keggdir | Slice | Keggundir | 3Droad | Song |
|---|---|---|---|---|---|---|---|---|---|---|
| SVGP | 0.313 | 0.380 | 0.294 | 0.186 | 0.662 | 0.089 | 0.131 | **0.122** | 0.511 | 0.797 |
| ODSVGP | 0.321 | **0.373** | **0.222** | 0.175 | 0.667 | 0.093 | 0.087 | **0.121** | 0.534 | 0.794 |
| DCSVGP | **0.156** | 0.379 | 0.286 | **0.150** | **0.604** | 0.086 | **0.039** | **0.121** | 0.434 | **0.777** |
| PPGPR | 0.306 | 0.392 | 0.377 | 0.282 | 0.659 | **0.091** | 0.205 | 0.125 | 0.552 | 0.780 |
| ODPPGPR | 0.333 | **0.376** | **0.277** | 0.394 | 0.647 | **0.090** | 0.092 | **0.123** | 0.565 | **0.778** |
| DCPPGPR | **0.178** | 0.395 | 0.348 | **0.226** | **0.632** | 0.089 | **0.042** | 0.124 | 0.543 | 0.779 |

| Test NLL | Pol | Elevators | Bike | Kin40k | Protein | Keggdir | Slice | Keggundir | 3Droad | Song |
|---|---|---|---|---|---|---|---|---|---|---|
| SVGP | 0.331 | 0.452 | 0.207 | -0.188 | 1.013 | -1.018 | -0.409 | -0.683 | 0.752 | 1.192 |
| ODSVGP | 0.278 | **0.433** | **-0.094** | -0.354 | 1.011 | **-1.029** | -0.578 | **-0.694** | 0.797 | 1.187 |
| DCSVGP | **-0.373** | 0.450 | 0.165 | **-0.502** | 0.919 | **-1.047** | **-1.293** | **-0.697** | 0.586 | **1.166** |
| PPGPR | -0.056 | **0.377** | -0.715 | -0.772 | **0.804** | -1.606 | -0.906 | -1.801 | 0.260 | 1.112 |
| ODPPGPR | -0.064 | **0.382** | -0.842 | -0.972 | **0.812** | -1.603 | -0.937 | -1.791 | 0.316 | 1.111 |
| DCPPGPR | **-0.588** | 0.403 | **-0.901** | **-1.067** | 0.854 | **-1.648** | **-1.574** | **-1.904** | **0.227** | 1.109 |

Table 3: Test RMSE and NLL on 10 regression datasets (lower is better). Results are averaged over 10 random train/validation/test splits. Statistical significance is indicated by **bold**. See the supplement for standard errors.

| Test RMSE | Pol | Elevators | Bike | Kin40k | Protein | Keggdir | Slice | Keggundir | 3Droad | Song |
|---|---|---|---|---|---|---|---|---|---|---|
| SVGP-DKL | 0.0614 | **0.343** | 0.013 | 0.067 | 0.598 | **0.0864** | **0.0189** | **0.120** | 0.329 | **0.773** |
| SVGP-DCDKL | **0.0513** | **0.343** | **0.011** | **0.047** | **0.577** | **0.0855** | 0.0223 | **0.119** | **0.288** | 0.774 |
| PPGPR-DKL | 0.0847 | **0.343** | 0.030 | 0.082 | **0.597** | **0.0871** | 0.0176 | 0.121 | 0.375 | 0.771 |
| PPGPR-DCDKL | **0.0593** | 0.346 | **0.027** | **0.061** | **0.594** | **0.0870** | **0.0171** | **0.120** | 0.350 | 0.771 |

| Test NLL | Pol | Elevators | Bike | Kin40k | Protein | Keggdir | Slice | Keggundir | 3Droad | Song |
|---|---|---|---|---|---|---|---|---|---|---|
| SVGP-DKL | -1.388 | **0.347** | -2.590 | -1.227 | 0.905 | **-1.045** | **-2.550** | **-0.712** | 0.320 | **1.161** |
| SVGP-DCDKL | **-1.600** | 0.348 | **-2.786** | **-1.637** | 0.888 | **-1.052** | -2.379 | **-0.713** | **0.173** | 1.164 |
| PPGPR-DKL | -2.324 | **0.303** | -3.096 | -1.739 | **0.712** | -1.627 | -2.625 | **-1.951** | -0.232 | **1.098** |
| PPGPR-DCDKL | **-2.584** | 0.358 | **-3.192** | **-2.186** | 1.194 | **-1.650** | **-2.676** | **-1.911** | **-0.340** | 1.117 |

**SVGP-DCDKL and PPGPR-DCDKL.** We then compare SVGP-DCDKL with baseline model SVGP-DKL [53] and comare PPGPR-DCDKL with baseline model PPGPR-DKL [22]. Here, all models learn inducing points in the input space rather than in the feature space and we refer to App. B.1.6 for more discussion on this choice and supporting results. Table 3 shows the test RMSE and NLL and we observe that SVGP-DCDKL yields better (or equivalent) RMSE and NLL on all datasets but the Slice dataset; PPGPR-DCDKL always yields better (or equivalent) RMSE and only obtains worse NLL on 3 datasets due to an overfitting issue similar to DCPPGPR.

## 5.2 Ablation Study On $\beta_2$

In Sec. 5.1, we evaluate decoupled models with fixed $\beta_2 = $ 1e-3 and observe mostly superior performance except for occasional overfitting with DCPPGPR and PPGPR-DCDKL. Here, we study how the regularization parameter $\beta_2$ affects model performance. We evaluate DCPPGPR with $\beta_2 \in \{0, 0.001, 0.005, 0.01, 0.05, 0.1, 0.5, 1.0\}$ on two datasets–Pol and Protein– that represent our general findings.

The first row of Fig. 2 shows performance on Pol: DCPPGPR gets lower RMSE and NLL with decreasing $\beta_2$, and $\beta_2 \to 0$ is desirable. We also see how the decoupled lengthscales $l_{\text{mean}}$ and $l_{\text{covar}}$ diverge from PPGPR's lengthscale as $\beta_2$ decreases. The second row of Fig. 2 shows results on Protein: DCPPGPR gets better RMSE but worse NLL as $\beta_2$ decreases and thus overfits when $\beta_2 < 0.1$. In this case, $\beta_2 = 0.1$ is the best. We refer to App. B.2 for more supporting results, and conclude that 1) small $\beta_2$ or even 0 is usually ideal and 2) adding back some small regularization $\beta_2 = 0.1$ appears to resolve the instances of overfitting we noticed.

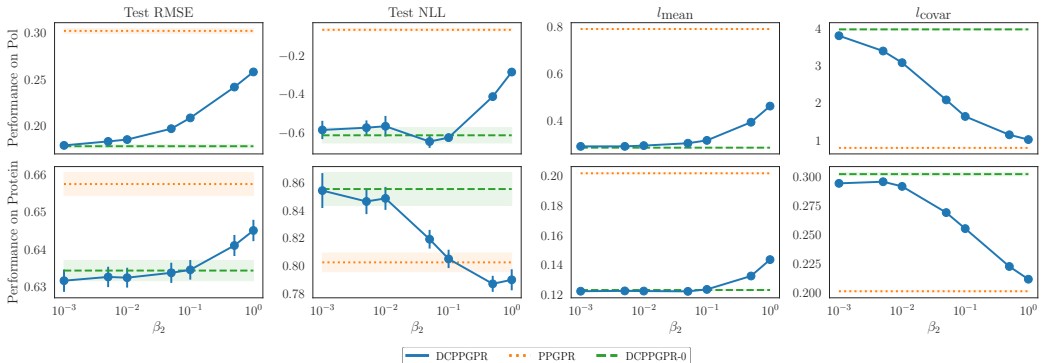

Figure 2: We evaluate DCPPGPR with varying $\beta_2$ on dataset Pol (**first row**) and Protein (**second row**). Results are averaged over 10 random dataset splits with standard errors included. Solid blue lines show DCPPGPR with nonzero $\beta_2$ and green dashed lines show DCPPGPR with $\beta_2 = 0$. Baseline PPGPR is shown in orange dotted line. **First two columns** contain test RMSE and NLL (lower is better). **Last two columns** show the learned decoupled lengthscales $l_{\mathrm{mean}}$ and $l_{\mathrm{covar}}$. PPGPR has equal $l_{\mathrm{mean}}$ and $l_{\mathrm{covar}}$ due to no decouling. See Sec. 5.2 for details.

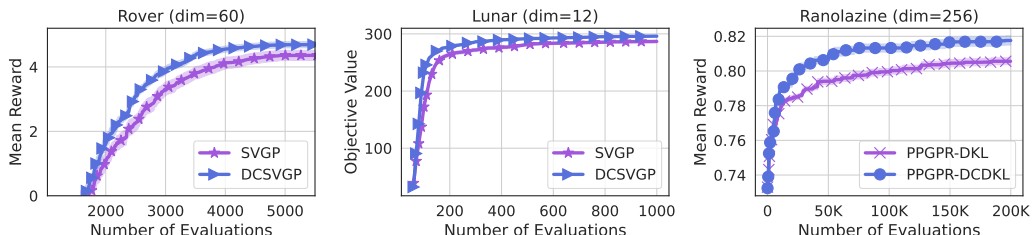

Figure 3: Optimization performance in terms of objective values on tasks Rover (dim=60), Lunar (dim=12) and Ranolazine (dim=256) is shown (higher is better). We compare baseline GP models (SVGP or PPGPR-DKL) with their decoupled counterparts (DCSVGP or PPGPR-DCDKL). Decoupling conditionals results in significant improvement on all tasks. See Sec. 5.3 for details.

## 5.3 Applications to Bayesian Optimization (BO)

In this section, we apply our decoupled variational GPs to BO tasks [13]. We consider 3 real BO tasks: the rover trajectory planning problem (Rover) [12, 52], the lunar landing reinforcement learning (Lunar) [12], and a challenging molecule design task Ranolazine MPO (Ranolazine) [5, 33]. We take TuRBO [12] as the BO algorithm and use different variational GPs in TuRBO. On Rover and Lunar, we use SVGP with TuRBO and compare SVGP with DCSVGP. On Ranolazine we first reduce the problem dimension to 256 using a Variational Autoencoder (VAE) [26], then perform BO in the latent space using PPGPR-DKL with TuRBO [33]. We compare PPGPR-DKL with PPGPR-DCDKL on Ranolazine. Consistent with our ablation study in Sec. 5.2, we find that $\beta_2 = 0$ performs the best on Rover and Ranolazine while Lunar requires more regularization with $\beta_2 = 0.1$. Fig. 3 summarizes optimization performance averaged on at least 10 runs – all decoupled models outperform their coupled counterparts, showing the advantage of decoupling conditionals in BO applications.

## 5.4 Plausible Extensions

We further explore extensions of decoupled conditionals to achieve more flexible predictive mean and variance of variational GPs. Beyond decoupling the same parametric form, we could model the predictive mean and covariance differently for a better fit. Despite lack of theoretical grounding, we evaluate two possibilities, both targeting a more complex predictive mean than the covariance.

**Replace the predictive mean of SVGP by a neural network.** In fact, the importance of the parametric form of the covariance field is mentioned in Jankowiak et al. [22]: "a good ansatz for the predictive variance is important for good uncertainty prediction". They suggest, but never implement

Table 4: Test NLL on 10 regression datasets (lower is better). Results are averaged over 10 random train/validation/test splits. Best ones with statistical significance are bold. See Sec. 5.4 for details.

| | Pol | Elevators | Bike | Kin40k | Protein | Keggdir | Slice | Keggundir | 3Droad | Song |
|---|---|---|---|---|---|---|---|---|---|---|
| SVGP | 0.331 | 0.452 | 0.207 | -0.188 | 1.013 | -1.018 | -0.409 | -0.683 | 0.752 | 1.192 |
| DCSVGP | -0.373 | 0.450 | 0.165 | -0.502 | 0.919 | **-1.047** | **-1.293** | -0.697 | **0.586** | **1.166** |
| NNSVGP | **-1.050** | 0.419 | **-1.231** | **-0.827** | 0.895 | **-1.060** | -0.948 | **-0.725** | 0.776 | 1.171 |
| PPGPR | -0.056 | **0.377** | -0.715 | -0.772 | 0.804 | -1.606 | -0.906 | -1.801 | 0.260 | 1.112 |
| DCPPGPR | -0.588 | 0.403 | -0.901 | -1.067 | 0.854 | **-1.648** | **-1.574** | **-1.904** | **0.227** | **1.109** |
| NNPPGPR | **-1.169** | 0.552 | **-1.560** | **-1.074** | **0.766** | -1.514 | -1.369 | -1.632 | 0.412 | 1.113 |

Table 5: Test RMSE and NLL on 10 regression datasets (lower is better). Results are averaged over 10 random train/validation/test splits. Best ones with statistical significance are bold. See Sec. 5.4 for details.

| **Test RMSE** | Pol | Elevators | Bike | Kin40k | Protein | Keggdir | Slice | Keggundir | 3Droad | Song |
|---|---|---|---|---|---|---|---|---|---|---|
| SVGP-DKL | 0.0614 | **0.343** | **0.013** | 0.067 | 0.598 | **0.0864** | **0.0189** | **0.120** | 0.329 | 0.773 |
| SVGP-DCDKL | **0.0513** | **0.343** | **0.011** | 0.047 | 0.577 | **0.0855** | 0.0223 | **0.119** | **0.288** | 0.774 |
| SVGP-MeanDKL | 0.0576 | **0.344** | 0.027 | **0.048** | **0.576** | **0.0859** | 0.0259 | **0.119** | **0.285** | **0.771** |
| PPGPR-DKL | 0.0847 | **0.343** | **0.030** | 0.082 | **0.597** | **0.0871** | **0.0176** | **0.121** | 0.375 | **0.771** |
| PPGPR-DCDKL | **0.0593** | 0.346 | 0.027 | 0.061 | **0.594** | **0.0870** | **0.0171** | **0.120** | 0.350 | **0.771** |
| PPGPR-MeanDKL | 0.0736 | **0.344** | 0.055 | **0.062** | **0.594** | **0.0868** | 0.0223 | **0.121** | 0.349 | **0.771** |

| **Test NLL** | Pol | Elevators | Bike | Kin40k | Protein | Keggdir | Slice | Keggundir | 3Droad | Song |
|---|---|---|---|---|---|---|---|---|---|---|
| SVGP-DKL | -1.388 | **0.347** | -2.590 | -1.227 | 0.905 | **-1.045** | **-2.550** | **-0.712** | 0.320 | **1.161** |
| SVGP-DCDKL | **-1.600** | **0.348** | **-2.786** | **-1.637** | 0.888 | **-1.052** | -2.379 | **-0.713** | 0.173 | 1.164 |
| SVGP-MeanDKL | -1.450 | **0.349** | -1.989 | **-1.645** | 0.883 | **-1.053** | -1.777 | **-0.715** | 0.163 | **1.160** |
| PPGPR-DKL | -2.324 | **0.303** | -3.096 | -1.739 | **0.712** | **-1.627** | -2.625 | **-1.951** | -0.232 | **1.098** |
| PPGPR-DCDKL | **-2.584** | 0.358 | **-3.192** | **-2.186** | 1.194 | **-1.650** | **-2.676** | **-1.911** | **-0.340** | 1.117 |
| PPGPR-MeanDKL | -2.362 | 0.312 | -2.831 | -2.150 | 1.048 | **-1.666** | -2.175 | **-1.906** | **-0.344** | 1.099 |

or evaluate, that one could replace the mean function with a neural network so that inducing points are only used for variance prediction. We empirically evaluate this idea, denoted as NNSVGP (and NNPPGPR). With a totally flexible predictive mean, NNSVGP and NNPPGPR yield best test RMSE over 9 out of 10 datasets (see supplement for the RMSE table). However, with the predictive mean fully independent of the GP framework, NNSVGP and NNPPGPR yield worse NLL typically. Table 4 reports test NLL, where results are generally mixed.

**Simplified deep kernels for covariances.** In the DKL setting, instead of using decoupled feature maps, one could use entirely different neural network architectures for the predictive mean and covariance. We empirically find that a simpler feature mapping, or even no feature mapping, for the covariance still make a plausible model with good performance but smaller training cost. We denote the method with no feature mapping for the covariance as SVGP-MeanDKL and PPGPR-MeanDKL.

We present results in Table 5. We observe SVGP-MeanDKL does not significantly outperform the decoupled models SVGP-DCDKL, but it improves baseline SVGP-DKL on 8 out of 10 datasets. PPGPR-MeanDKL performs similarly in terms of the RMSE metric (better than baseline PPGPR-DKL but worse than PPGPR-DCDKL). However, in terms of NLL, it gives poor uncertainty prediction (worse NLL). This suggests that it is more beneficial to have different models (decoupled feature extractors in SVGP-DCDKL) for conditional mean and covariance rather than only modeling the conditional mean (SVGP-MeanDKL) or having same models (SVGP-DKL), and SVGP-based model is more robust than the PPGPR-based model in uncertainty prediction.

# 6   Conclusion

Variational GPs scale approximate GP inference to millions of training examples but may yield suboptimal accuracy due to lack of model expressiveness or poor model fitting. We propose a simple idea to improve model expressiveness by decoupling the parametric form of the mean and variance in the conditionals of variational GPs. We derive an ELBO for our model with decoupled conditionals, which end up being similar to the ELBO of standard variational GP with an additional a regularization

term. Our method is simple yet effective, and it applies to various variational GPs. We provide two concrete examples, one decoupling kernel lengthscales in the basic variational GP setting and one decoupling the feature mapping in the deep kernel learning setting. Through extensive empirical study, we show that the decoupled conditionals effectively improve the accuracy of variational GPs in terms of both mean and uncertainty prediction. We also empirically study two plausible extensions of our method, motivated by the idea of modeling the mean and variance differently, but we conclude that they are not as effective as our decoupled models. Current limitations of our work and therefore future directions include but not limit to: 1) the application of decoupled conditionals to more SVGP-based models; 2) further generalization of flexible forms of training and testing conditionals that improve model performance; 3) the application of decoupled variational GP framework to tasks other than regression or BO, such as classification tasks.

## 7  Acknowledgements

We acknowledge support from Simons Foundation. This work was supported by a grant from the Simons Foundation (601956, DSB). JRG is supported by NSF award IIS-2145644.

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

# A Methodology

## A.1 The Evidence Lower Bound (ELBO)

Here we privide detailed derivation of the evidence lower bound (ELBO) for the proposed decoupled SVGP model fitting. The goal of variational learning remains the same:

$$\min \mathrm{KL}(q(\mathbf{f}, \mathbf{f}_m)||p(\mathbf{f}, \mathbf{f}_m|\mathbf{y})) = \min \left( \mathbb{E}[\log(q(\mathbf{f}, \mathbf{f}_m))] - \mathbb{E}[\log(p(\mathbf{f}, \mathbf{f}_m, \mathbf{y}))] + \log(p(\mathbf{y})) \right),$$

where in the standard case $q(\mathbf{f}, \mathbf{f}_m) = p(\mathbf{f}|\mathbf{f}_m)\phi(\mathbf{f}_m)$, but we further approximate the training conditional $p(\mathbf{f}|\mathbf{f}_m)$ by a decoupled one $\psi(\mathbf{f}|\mathbf{f}_m)$ and have $q(\mathbf{f}, \mathbf{f}_m) = \psi(\mathbf{f}|\mathbf{f}_m)\phi(\mathbf{f}_m)$. Following the standard variational inference [2], we have the ELBO for decoupled SVGP:

$$\begin{aligned}
\log(p(\mathbf{y})) \geq \mathrm{ELBO}(q) &= \mathbb{E}[\log(p(\mathbf{f}, \mathbf{f}_m, \mathbf{y}))] - \mathbb{E}[\log(q(\mathbf{f}, \mathbf{f}_m))] \\
&= \mathbb{E}[\log(p(\mathbf{y}|\mathbf{f}, \mathbf{f}_m))] - \mathrm{KL}(q(\mathbf{f}, \mathbf{f}_m)||p(\mathbf{f}, \mathbf{f}_m)).
\end{aligned} \tag{7}$$

The first likelihood term in Eq. 7 is

$$\begin{aligned}
\mathbb{E}[\log(p(\mathbf{y}|\mathbf{f}, \mathbf{f}_m))] &= \int \int \log(p(\mathbf{y}|\mathbf{f}, \mathbf{f}_m)) q(\mathbf{f}, \mathbf{f}_m) \, d\mathbf{f} d\mathbf{f}_m \\
&= \int \phi(\mathbf{f}_m) \left( \int \log(p(\mathbf{y}|\mathbf{f}, \mathbf{f}_m)) \psi(\mathbf{f}|\mathbf{f}_m) \, d\mathbf{f} \right) d\mathbf{f}_m \\
&= \int \phi(\mathbf{f}_m) \left( \int \log(p(\mathbf{y}|\mathbf{f})) \psi(\mathbf{f}|\mathbf{f}_m) \, d\mathbf{f} \right) d\mathbf{f}_m \\
&= \sum_{i=1}^{n} \left\{ \log \mathcal{N}(y_i | \tilde{\mu}_{\mathbf{f}}(\mathbf{x}_i), \sigma^2) - \frac{\tilde{\sigma}_{\mathbf{f}}(\mathbf{x}_i)^2}{2\sigma^2} \right\},
\end{aligned}$$

where the predictive mean $\tilde{\mu}_{\mathbf{f}}(\mathbf{x}_i)$ and latent function variance $\tilde{\sigma}_{\mathbf{f}}(\mathbf{x}_i)^2$ are

$$\tilde{\mu}_{\mathbf{f}}(\mathbf{x}_i) = \mathbf{Q}_{im}\mathbf{Q}_{mm}^{-1}\mathbf{m}, \quad \tilde{\sigma}_{\mathbf{f}}(\mathbf{x}_i)^2 = \tilde{\mathbf{K}}_{ii} + \mathbf{Q}_{im}\mathbf{Q}_{mm}^{-1}\mathbf{S}\mathbf{Q}_{mm}^{-1}\mathbf{Q}_{mi}.$$

The second KL divergence term in Eq. 7 is

$$\begin{aligned}
\mathrm{KL}(q(\mathbf{f}, \mathbf{f}_m)||p(\mathbf{f}, \mathbf{f}_m)) &= \int \psi(\mathbf{f}|\mathbf{f}_m)\phi(\mathbf{f}_m) \log \frac{\psi(\mathbf{f}|\mathbf{f}_m)\phi(\mathbf{f}_m)}{p(\mathbf{f}|\mathbf{f}_m)p(\mathbf{f}_m)} \, d\mathbf{f} d\mathbf{f}_m \\
&= \int \psi(\mathbf{f}|\mathbf{f}_m)\phi(\mathbf{f}_m) \log \frac{\psi(\mathbf{f}|\mathbf{f}_m)}{p(\mathbf{f}|\mathbf{f}_m)} \, d\mathbf{f} d\mathbf{f}_m + \int \psi(\mathbf{f}|\mathbf{f}_m)\phi(\mathbf{f}_m) \log \frac{\phi(\mathbf{f}_m)}{p(\mathbf{f}_m)} \, d\mathbf{f} d\mathbf{f}_m \\
&= \int \phi(\mathbf{f}_m)\mathrm{KL}(\psi(\mathbf{f}|\mathbf{f}_m)||p(\mathbf{f}|\mathbf{f}_m)) \, d\mathbf{f}_m + \mathrm{KL}(\phi(\mathbf{f}_m)||p(\mathbf{f}_m)) \\
&= \underbrace{\mathbb{E}_{\phi(\mathbf{f}_m)}[\mathrm{KL}(\psi(\mathbf{f}|\mathbf{f}_m)||p(\mathbf{f}|\mathbf{f}_m))]}_{:=\Omega} + \mathrm{KL}(\phi(\mathbf{f}_m)||p(\mathbf{f}_m)).
\end{aligned}$$

The KL divergence term is almost the same as that in the standard ELBO, with an additional term $\Omega := \int \phi(\mathbf{f}_m)\mathrm{KL}(\psi(\mathbf{f}|\mathbf{f}_m)||p(\mathbf{f}|\mathbf{f}_m)) \, d\mathbf{f}_m$.

**Analytic expression of $\Omega$**   We could compute explicitly the analytical expression for the $\Omega$ term. For notational simplicity, let decoupled and exact training conditionals be

$$\begin{aligned}
\psi(\mathbf{f}|\mathbf{f}_m) &= \mathcal{N}(\mathbf{Q}_{nm}\mathbf{Q}_{mm}^{-1}\mathbf{f}_m, \tilde{\mathbf{K}}_{nn}) := \mathcal{N}(\boldsymbol{\mu}_1, \boldsymbol{\Sigma}), \\
p(\mathbf{f}|\mathbf{f}_m) &= \mathcal{N}(\mathbf{K}_{nm}\mathbf{K}_{mm}^{-1}\mathbf{f}_m, \tilde{\mathbf{K}}_{nn}) := \mathcal{N}(\boldsymbol{\mu}_2, \boldsymbol{\Sigma}).
\end{aligned}$$

The $\Omega$ term is an expectation of a quadratic form:

$$\begin{aligned}
\Omega &= \int \phi(\mathbf{f}_m)\mathrm{KL}(\psi(\mathbf{f}|\mathbf{f}_m)||p(\mathbf{f}|\mathbf{f}_m)) \, d\mathbf{f}_m \\
&= \frac{1}{2} \int \phi(\mathbf{f}_m) \left( (\boldsymbol{\mu}_1 - \boldsymbol{\mu}_2)^T \boldsymbol{\Sigma}^{-1} (\boldsymbol{\mu}_1 - \boldsymbol{\mu}_2) \right) \, d\mathbf{f}_m \\
&= \frac{1}{2} \mathbb{E}[\mathbf{f}_m^T \mathbf{T} \mathbf{f}_m] = \frac{1}{2} \left( \mathrm{Tr}(\mathbf{T}\mathbf{S}) + \mathbf{m}^T \mathbf{T} \mathbf{m} \right),
\end{aligned} \tag{8}$$

where $\mathbf{T} = \mathbf{A}^T \boldsymbol{\Sigma}^{-1} \mathbf{A}$, $\mathbf{A} = \mathbf{Q}_{nm}\mathbf{Q}_{mm}^{-1} - \mathbf{K}_{nm}\mathbf{K}_{mm}^{-1}$, and $\phi(\mathbf{f}_m) = \mathcal{N}(\mathbf{m}, \mathbf{S})$.

## A.2 Whitening

In this section, we provide detailed discussion of the whitening scheme used in the decoupled models. We first present the standard whitening scheme used in variational GP models, then derive two plausible whitening schemes for decoupled models – the $\mathbf{K}$-whitening and the $\mathbf{Q}$-whitening. We implemented both ways of whitening for decoupled models, and empirically found that the latter ($\mathbf{Q}$-whitening) performs better (See Section B.1.5).

### A.2.1  The standard whitening

For standard models like SVGP, using the square root of $\mathbf{K}_{mm}$, the variational distribution is "whitened" (reparameterized) in the following way:

$$\bar{q}(\mathbf{f}_m) = \mathcal{N}(\bar{\mathbf{m}}, \bar{\mathbf{S}}) = \mathcal{N}(\mathbf{K}_{mm}^{-1/2}\mathbf{m}, \mathbf{K}_{mm}^{-1/2}\mathbf{S}\mathbf{K}_{mm}^{-1/2}).$$

Under such reparameterization, the KL divergence is equivalent to a simpler form:

$$
\begin{aligned}
\mathrm{KL}(q(\mathbf{f}_m)||p(\mathbf{f}_m)) &= \frac{1}{2}\left[-\log|\mathbf{S}| + \log|\mathbf{K}_{mm}| + \mathrm{Tr}(\mathbf{K}_{mm}^{-1}\mathbf{S}) + \mathbf{m}^T\mathbf{K}_{mm}^{-1}\mathbf{m} - M\right] \\
&= \frac{1}{2}\left[-\log|\bar{\mathbf{S}}| + \mathrm{Tr}(\mathbf{K}_{mm}^{-1/2}\mathbf{K}_{mm}^{-1/2}\mathbf{K}_{mm}^{1/2}\bar{\mathbf{S}}\mathbf{K}_{mm}^{1/2}) + \bar{\mathbf{m}}^T\mathbf{K}_{mm}^{1/2}\mathbf{K}_{mm}^{-1/2}\mathbf{K}_{mm}^{-1/2}\mathbf{K}_{mm}^{1/2}\bar{\mathbf{m}} - M\right] \\
&= \frac{1}{2}\left[-\log|\bar{\mathbf{S}}| + \mathrm{Tr}(\bar{\mathbf{S}}) + \bar{\mathbf{m}}^T\bar{\mathbf{m}} - M\right] \\
&= \mathrm{KL}(\bar{q}(\mathbf{f}_m)||p_0(\mathbf{f}_m)),
\end{aligned}
$$

where $p_0(\mathbf{f}_m) = \mathcal{N}(\mathbf{0}, \mathbf{I})$ is the standard Normal, and $M$ is the number of inducing points. Now plug $\bar{\mathbf{m}}$ and $\bar{\mathbf{S}}$ in the predictive distribution $p(\mathbf{f})$, we have the predictive mean $\mu_{\mathbf{f}}(\mathbf{x}_i)$ and latent function variance $\sigma_{\mathbf{f}}(\mathbf{x}_i)^2$ in simplified forms as well:

$$
\begin{aligned}
\mu_{\mathbf{f}}(\mathbf{x}_i) &= \mathbf{K}_{im}\mathbf{K}_{mm}^{-1/2}\bar{\mathbf{m}}, \\
\sigma_{\mathbf{f}}(\mathbf{x}_i)^2 &= \mathbf{K}_{ii} - \mathbf{K}_{im}\mathbf{K}_{mm}^{-1}\mathbf{K}_{mi} + \mathbf{K}_{im}\mathbf{K}_{mm}^{-1/2}\bar{\mathbf{S}}\mathbf{K}_{mm}^{-1/2}\mathbf{K}_{mi}.
\end{aligned}
$$

### A.2.2  The K-whitening

For decoupled models like DCSVGP, we could similarly use the square root of $\mathbf{K}_{mm}$ to whiten the variational distribution:

$$\bar{q}(\mathbf{f}_m) = \mathcal{N}(\bar{\mathbf{m}}, \bar{\mathbf{S}}) = \mathcal{N}(\mathbf{K}_{mm}^{-1/2}\mathbf{m}, \mathbf{K}_{mm}^{-1/2}\mathbf{S}\mathbf{K}_{mm}^{-1/2}).$$

Same as the standard whitening, the KL divergence is simplified:

$$
\begin{aligned}
\mathrm{KL}(q(\mathbf{f}_m)||p(\mathbf{f}_m)) &= \frac{1}{2}\left[-\log|\mathbf{S}| + \log|\mathbf{K}_{mm}| + \mathrm{Tr}(\mathbf{K}_{mm}^{-1}\mathbf{S}) + \mathbf{m}^T\mathbf{K}_{mm}^{-1}\mathbf{m} - M\right] \\
&= \frac{1}{2}\left[-\log|\bar{\mathbf{S}}| + \mathrm{Tr}(\bar{\mathbf{S}}) + \bar{\mathbf{m}}^T\bar{\mathbf{m}} - M\right] \\
&= \mathrm{KL}(\bar{q}(\mathbf{f}_m)||p_0(\mathbf{f}_m)),
\end{aligned}
$$

where $p_0(\mathbf{f}_m) = \mathcal{N}(\mathbf{0}, \mathbf{I})$ is the standard Normal, and $M$ is the number of inducing points. However, if we plug $\bar{\mathbf{m}}$ and $\bar{\mathbf{S}}$ in the predictive distribution $p(\mathbf{f})$, the predictive mean $\tilde{\mu}_{\mathbf{f}}(\mathbf{x}_i)$ and latent function variance $\tilde{\sigma}_{\mathbf{f}}(\mathbf{x}_i)^2$ are not simplified in the same way as the standard case:

$$
\begin{aligned}
\tilde{\mu}_{\mathbf{f}}(\mathbf{x}_i) &= \mathbf{Q}_{im}\mathbf{Q}_{mm}^{-1}\mathbf{K}_{mm}^{1/2}\bar{\mathbf{m}}, \\
\tilde{\sigma}_{\mathbf{f}}(\mathbf{x}_i)^2 &= \mathbf{K}_{ii} - \mathbf{K}_{im}\mathbf{K}_{mm}^{-1}\mathbf{K}_{mi} + \mathbf{Q}_{im}\mathbf{Q}_{mm}^{-1}\mathbf{K}_{mm}^{1/2}\bar{\mathbf{S}}\mathbf{K}_{mm}^{1/2}\mathbf{Q}_{mm}^{-1}\mathbf{Q}_{mi}.
\end{aligned}
$$

### A.2.3  The Q-whitening

For decoupled models like DCSVGP, another plausible way of whitening for decoupled models is to use the square root of $\mathbf{Q}_{mm}$:

$$\bar{q}(\mathbf{f}_m) = \mathcal{N}(\bar{\mathbf{m}}, \bar{\mathbf{S}}) = \mathcal{N}(\mathbf{Q}_{mm}^{-1/2}\mathbf{m}, \mathbf{Q}_{mm}^{-1/2}\mathbf{S}\mathbf{Q}_{mm}^{-1/2}).$$

In this way, the KL divergence term is equivalent to a different form:

$$\mathrm{KL}(q(\mathbf{f}_m)\|p(\mathbf{f}_m)) = \frac{1}{2}\left[-\log|\mathbf{S}| + \log|\mathbf{K}_{mm}| + \mathrm{Tr}(\mathbf{K}_{mm}^{-1}\mathbf{S}) + \mathbf{m}^T\mathbf{K}_{mm}^{-1}\mathbf{m} - M\right]$$

$$=\frac{1}{2}\left[-\log|\mathbf{Q}_{mm}^{1/2}\bar{\mathbf{S}}\mathbf{Q}_{mm}^{1/2}| + \log|\mathbf{K}_{mm}| + \mathrm{Tr}(\mathbf{K}_{mm}^{-1}\mathbf{Q}_{mm}^{1/2}\bar{\mathbf{S}}\mathbf{Q}_{mm}^{1/2}) + \bar{\mathbf{m}}^T\mathbf{Q}_{mm}^{1/2}\mathbf{K}_{mm}^{-1}\mathbf{Q}_{mm}^{1/2}\bar{\mathbf{m}} - M\right]$$

$$=\frac{1}{2}\left[-\log|\mathbf{Q}_{mm}^{1/2}\bar{\mathbf{S}}\mathbf{Q}_{mm}^{1/2}| + \log|\mathbf{K}_{mm}^{1/2}\mathbf{K}_{mm}^{1/2}| + \mathrm{Tr}(\mathbf{K}_{mm}^{-1/2}\mathbf{K}_{mm}^{-1/2}\mathbf{Q}_{mm}^{1/2}\bar{\mathbf{S}}\mathbf{Q}_{mm}^{1/2}) + \bar{\mathbf{m}}^T\mathbf{Q}_{mm}^{1/2}\mathbf{K}_{mm}^{-1/2}\mathbf{K}_{mm}^{-1/2}\mathbf{Q}_{mm}^{1/2}\bar{\mathbf{m}} - M\right]$$

$$=\frac{1}{2}\left[-\log|\bar{\mathbf{S}}| + \log|\mathbf{Q}_{mm}^{-1/2}\mathbf{K}_{mm}^{1/2}\mathbf{K}_{mm}^{1/2}\mathbf{Q}_{mm}^{-1/2}| + \mathrm{Tr}(\mathbf{K}_{mm}^{-1/2}\mathbf{Q}_{mm}^{1/2}\bar{\mathbf{S}}\mathbf{Q}_{mm}^{1/2}\mathbf{K}_{mm}^{-1/2}) + \bar{\mathbf{m}}^T\mathbf{Q}_{mm}^{1/2}\mathbf{K}_{mm}^{-1/2}\mathbf{K}_{mm}^{-1/2}\mathbf{Q}_{mm}^{1/2}\bar{\mathbf{m}} - M\right]$$

$$=\frac{1}{2}\left[-\log|\bar{\mathbf{S}}| + \log|\bar{\mathbf{L}}\bar{\mathbf{L}}^T| + \mathrm{Tr}(\bar{\mathbf{L}}^{-1}\bar{\mathbf{S}}\bar{\mathbf{L}}^{-T}) + \bar{\mathbf{m}}^T\bar{\mathbf{L}}^{-T}\bar{\mathbf{L}}^{-1}\bar{\mathbf{m}} - M\right]$$

$$=\frac{1}{2}\left[-\log|\bar{\mathbf{S}}| + \log|\bar{\mathbf{K}}| + \mathrm{Tr}(\bar{\mathbf{K}}^{-1}\bar{\mathbf{S}}) + \bar{\mathbf{m}}^T\bar{\mathbf{K}}^{-1}\bar{\mathbf{m}} - M\right]$$

$$=\mathrm{KL}(\bar{q}(\mathbf{f}_m)\|\bar{p}(\mathbf{f}_m)),$$

where $\bar{p}(\mathbf{f}_m) = \mathcal{N}(\mathbf{0}, \bar{\mathbf{K}})$ and $\bar{\mathbf{K}} = \bar{\mathbf{L}}\bar{\mathbf{L}}^T$, $\bar{\mathbf{L}} = \mathbf{Q}_{mm}^{-1/2}\mathbf{K}_{mm}^{1/2}$ and $M$ is the number of inducing points. Now plug $\bar{\mathbf{m}}$ and $\bar{\mathbf{S}}$ in the predictive distribution $p(\mathbf{f})$, we have the predictive mean $\tilde{\mu}_{\mathbf{f}}(\mathbf{x}_i)$ and latent function variance $\tilde{\sigma}_{\mathbf{f}}(\mathbf{x}_i)^2$ simplified in a similar way as the standard case:

$$\tilde{\mu}_{\mathbf{f}}(\mathbf{x}_i) = \mathbf{Q}_{im}\mathbf{Q}_{mm}^{-1/2}\bar{\mathbf{m}},$$
$$\tilde{\sigma}_{\mathbf{f}}(\mathbf{x}_i)^2 = \mathbf{K}_{ii} - \mathbf{K}_{im}\mathbf{K}_{mm}^{-1}\mathbf{K}_{mi} + \mathbf{Q}_{im}\mathbf{Q}_{mm}^{-1/2}\bar{\mathbf{S}}\mathbf{Q}_{mm}^{-1/2}\mathbf{Q}_{mi}.$$

As discussed in the main paper, it is not surprising that the $\mathbf{Q}$-whitening performs better in practice. This is because the $\mathbf{K}$-whitening simplifies the KL divergence but leaves the predictive distribution complicated, while the $\mathbf{Q}$-whitening does the opposite, and a complicated predictive distribution makes the optimization much harder than a complicated KL divergence term.

## B Experiments

### B.1 Regression Tasks

**Experiment setup** Table 6 provides details on the 10 UCI regression datasets [10] and the number of inducing points used for each dataset. For ODSVGP, since the goal is to use more inducing points for mean and less for covariance under the same budget, we use the setup in the original paper [41]: if SVGP uses 4X inducing points, then ODSVGP uses 7X inducing points for mean and 3X inducing points for covariance. We use the Adam [25] optimizer with a multistep scheduler to train all models on all datasets, and we train for 300 epochs using training batch size 1024. We selected the best training hyperparameters for SVGP and use the same ones for all models – learning rate lr $= 5$e-3 and a multistep learning rate scheduler (multiplicative factor $\gamma = 0.2$).

#### B.1.1 Learned lengthscales

Table 6 also provides learned lengthscales of all models, averaged over 10 random dataset splits, as supplementary results for Table 2. As is shown in Table 6, we observe how lengthscales are decoupled in DCSVGP and DCPPGPR – the lengthscale for mean function $l_{\mathrm{mean}}$ is typically smaller than the shared lengthscale $l$, while the lengthscale for covariance function $l_{\mathrm{covar}}$ is typically larger than the shared lengthscale $l$. This is indeed consistent with the motivation of decoupling lengthscales since a smaller lengthscale for the mean function would typically lead to a better mean fit when the number of inducing points is limited compare to total training data; when the mean function is well fit, the lengthscale for the covariance can be larger to further reduce the predictive covariance.

#### B.1.2 Error bars of main results

Here we provide standard errors of main results Table 2 and Table 3 – see Figure 4 and Figure 5 respectively. Error bars for other similar results, i.e. Tables in the Appendix, are similarly small and not informative and therefore we avoid redundant results.

Table 6: We present **details on UCI datasets** and number of inducing points used in SVGP, DCSVGP, PPGPR and DCPPGPR here. We also compare the **learned lengthscale** values (averaged over 10 random dataset splits): lengthscale $l$ of baseline models SVGP, ODSVGP, PPGPR and ODPPGPR, lengthscales $l_{\text{mean}}$ and $l_{\text{covar}}$ of decoupled models DCSVGP and DCPPGPR. We observe that the lengthscale for mean $l_{\text{mean}}$ is typically smaller than the shared lengthscale $l$, while the lengthscale for covariance $l_{\text{covar}}$ is typically larger than the shared lengthscale $l$. See App. B.1 and App. B.1.1 for more details.

| | Pol | Elevators | Bike | Kin40k | Protein | Keggdir | Slice | Keggundir | 3Droad | Song |
|---|---|---|---|---|---|---|---|---|---|---|
| Total number of data | 11250 | 12449 | 13034 | 30000 | 34297 | 36620 | 40125 | 47706 | 326155 | 386508 |
| Input dimension | 26 | 18 | 17 | 8 | 9 | 20 | 380 | 26 | 3 | 90 |
| #inducing points | 500 | 500 | 500 | 800 | 800 | 800 | 800 | 800 | 1000 | 1000 |
| SVGP $l$ | 0.976 | 1.370 | 2.330 | 1.220 | 0.260 | 1.150 | 19.600 | 2.270 | 0.120 | 1.110 |
| ODSVGP $l$ | 1.418 | 1.561 | 3.162 | 1.710 | 0.412 | 1.726 | 20.694 | 2.665 | 0.158 | 1.664 |
| DCSVGP $l_{\text{mean}}$ | 0.291 | 0.761 | 1.174 | 0.673 | 0.104 | 0.791 | 4.956 | 1.273 | 0.038 | 0.346 |
| DCSVGP $l_{\text{covar}}$ | 3.304 | 2.257 | 3.459 | 1.721 | 0.555 | 3.190 | 28.112 | 5.876 | 0.170 | 2.124 |
| PPGPR $l$ | 0.789 | 1.060 | 2.890 | 1.300 | 0.200 | 1.030 | 21.000 | 2.510 | 0.110 | 0.490 |
| ODPPGPR $l$ | 1.079 | 1.171 | 3.387 | 1.825 | 0.261 | 1.298 | 20.536 | 2.583 | 0.137 | 0.591 |
| DCPPGPR $l_{\text{mean}}$ | 0.290 | 0.790 | 1.423 | 0.797 | 0.122 | 0.923 | 5.053 | 2.161 | 0.100 | 0.349 |
| DCPPGPR $l_{\text{covar}}$ | 3.811 | 1.361 | 4.556 | 1.793 | 0.298 | 2.122 | 26.734 | 4.814 | 0.138 | 0.640 |

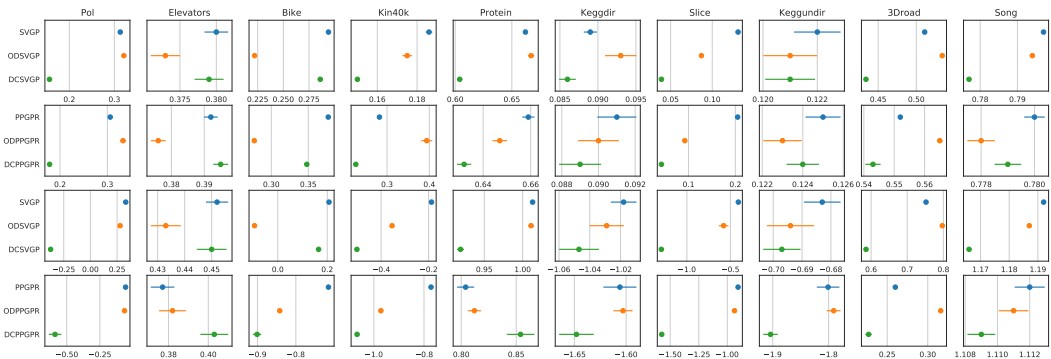

Figure 4: **Test RMSE and NLL results with standard errors** on UCI datasets, same results as Table 2 in main paper. We **compare DCSVGP to SVGP and ODSVGP** and **compare DCPPGPR to PPGPR and ODPPGPR**. Row 1 and 2 include RMSE results and row 3 and 4 include NLL results (same row order as Table 2 in the main paper). See App. B.1.2 for more details.

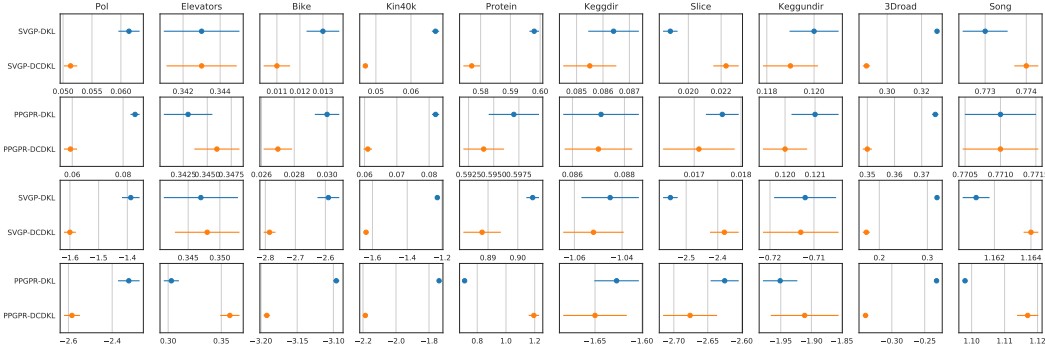

Figure 5: **Test RMSE and NLL results with standard errors** on UCI datasets, same results as Table 3 in main paper. We **compare SVGP-DCDKL to SVGP-DKL** and **compare PPGPR-DCDKL to PPGPR-DKL**. Row 1 and 2 include RMSE results and row 3 and 4 include NLL results (same row order as Table 3 in the main paper). See App. B.1.2 for more details.

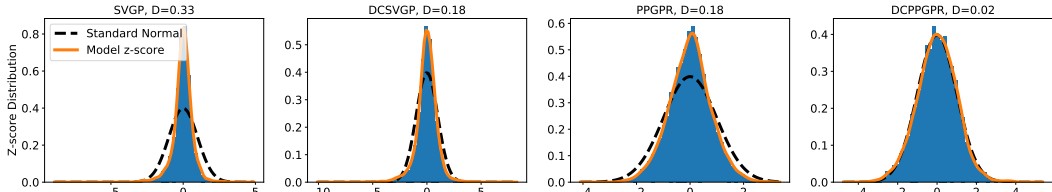

Figure 6: On dataset Kin40K, we plot **z-scores distributions** of model prediction (orange curves), and compare with the standard Normal (black dashed curves). The closer the model predictive distribution is to the standard Normal, the better the model is calibrated. Clearly, **decoupled models DCSVGP and DCPPGPR are better calibrated** compare to their counterparts SVGP and PPGPR, respectively. Quantitatively, in the titles we show the **Wasserstein distance** $D$ between the two distributions, the smaller the better. See App. B.1.3 for more details.

Table 7: Test RMSE and NLL on 10 univariate regression datasets (lower is better) with **ARD kernels**. We compare decoupled models DCSVGP and DCPPGPR with their coupled counterparts SVGP and PPGPR. Results are averaged over 6 random train/validation/test splits. Best ones with statistical significance are bold. See App. B.1.4 for more details.

| Test RMSE | Pol | Elevators | Bike | Kin40k | Protein | Keggdir | Slice | Keggundir | 3Droad | Song |
|---|---|---|---|---|---|---|---|---|---|---|
| SVGP | 0.167 | **0.370** | 0.089 | 0.166 | 0.646 | **0.089** | 0.111 | **0.122** | 0.376 | 0.794 |
| DCSVGP | **0.078** | **0.366** | **0.058** | **0.116** | **0.590** | **0.087** | **0.041** | **0.120** | **0.323** | **0.772** |
| PPGPR | 0.166 | **0.376** | 0.127 | 0.240 | 0.643 | **0.092** | 0.150 | **0.124** | **0.450** | 0.779 |
| DCPPGPR | **0.084** | **0.372** | **0.064** | **0.163** | **0.614** | **0.090** | **0.047** | **0.123** | **0.458** | **0.776** |

| Test NLL | Pol | Elevators | Bike | Kin40k | Protein | Keggdir | Slice | Keggundir | 3Droad | Song |
|---|---|---|---|---|---|---|---|---|---|---|
| SVGP | -0.286 | 0.426 | -0.873 | -0.315 | 0.991 | **-1.027** | -0.482 | -0.689 | 0.457 | 1.188 |
| DCSVGP | **-1.102** | **0.414** | **-1.453** | **-0.712** | **0.901** | **-1.038** | **-1.279** | **-0.710** | **0.289** | **1.161** |
| PPGPR | -0.871 | **0.359** | -1.818 | -0.844 | **0.783** | **-1.603** | -0.919 | -1.900 | **-0.084** | 1.108 |
| DCPPGPR | **-1.494** | 0.392 | **-2.223** | **-1.177** | 0.885 | **-1.622** | **-1.505** | **-1.989** | **-0.083** | 1.121 |

### B.1.3  Model Calibration

To study how decoupled conditionals affect model calibration, we evaluate the z-score distribution from model predictions on all UCI datasets and compare it to the standard Normal. The closer the z-score distribution from model predictions is to the standard Normal, the better the model is calibrated. As an example, Figure 6 shows the z-score distributions of SVGP, DCSVGP, PPGPR and DCPPGPR on the Kin40k dataset. As Figure 6 shows, on the Kin40k dataset, **decoupled models DCSVGP and DCPPGPR are better calibrated** compare to their counterparts SVGP and PPGPR, respectively. Quantitatively, we compute the Wasserstein distance $D$ [37] between the model z-score distribution and standard Normal (see subtitles in Figure 6 for example), the smaller the better. Averaged over all 10 UCI datasets, SVGP and DCSVGP both have $D = 0.27$, PPGPR has $D = 0.22$ and DCPPGPR has the best $D = 0.17$. Therefore, we conclude that decoupling conditionals does not lead to a poorly calibrated model, but can even improve model calibration on average.

### B.1.4  ARD kernels

Here we provide additional results on regression tasks using **ARD kernels**. In Table 2, we show that decoupled models outperform the baseline models using non-ARD kernels (kernels with one-dimensional lengthscale). Here, we use ARD kernels (kernels with multi-dimensional lengthscales, one for each input dimension) for all models and **show consistent advantages of using decoupled models**. Table 7 summarizes results, similar to the main results using non-ARD kernels (Table 2 in main paper), we observe that all decoupled models outperform baseline models.

### B.1.5  Whitening of decoupled models

In Section A.2, we discussed two plausible whitening schemes for the decoupled models and claimed that the **Q**-whitening is more favorable than the **K**-whitening due to the simplified predictive

Table 8: Test RMSE and NLL on 10 univariate regression datasets (lower is better). **Using the DCSVGP model, we compare the K-whitening and Q-whitening** discussed in App. A.2. Results are averaged over 10 random train/validation/test splits. Best ones with statistical significance are bold. See App. B.1.5 for more details.

| **Test RMSE** | Pol | Elevators | Bike | Kin40k | Protein | Keggdir | Slice | Keggundir | 3Droad | Song |
|---|---|---|---|---|---|---|---|---|---|---|
| K-whitening | 0.163 | **0.380** | **0.260** | 0.158 | 0.635 | **0.087** | **0.041** | **0.119** | 0.437 | **0.774** |
| Q-whitening | **0.156** | 0.379 | 0.286 | **0.150** | **0.604** | 0.086 | 0.039 | 0.121 | 0.434 | 0.777 |

| **Test NLL** | Pol | Elevators | Bike | Kin40k | Protein | Keggdir | Slice | Keggundir | 3Droad | Song |
|---|---|---|---|---|---|---|---|---|---|---|
| K-whitening | -0.099 | 0.457 | **0.083** | -0.453 | 0.966 | **-1.045** | -1.243 | **-0.703** | 0.591 | **1.163** |
| Q-whitening | **-0.373** | 0.450 | 0.165 | **-0.502** | 0.919 | -1.047 | -1.293 | -0.697 | 0.586 | 1.166 |

Table 9: Test RMSE and NLL on 10 univariate regression datasets (lower is better). **Using the DCPPGPR model, we compare the K-whitening and Q-whitening** discussed in App. A.2. Results are averaged over 10 random train/validation/test splits. Best ones with statistical significance are bold. See App. B.1.5 for more details.

| **Test RMSE** | Pol | Elevators | Bike | Kin40k | Protein | Keggdir | Slice | Keggundir | 3Droad | Song |
|---|---|---|---|---|---|---|---|---|---|---|
| K-whitening | **0.178** | **0.387** | **0.296** | **0.214** | 0.659 | **0.089** | 0.046 | **0.123** | 0.567 | 0.781 |
| Q-whitening | **0.178** | 0.395 | 0.348 | 0.226 | **0.632** | 0.089 | 0.042 | 0.124 | 0.543 | 0.779 |

| **Test NLL** | Pol | Elevators | Bike | Kin40k | Protein | Keggdir | Slice | Keggundir | 3Droad | Song |
|---|---|---|---|---|---|---|---|---|---|---|
| K-whitening | 0.434 | 0.475 | -0.786 | -0.941 | **0.813** | -1.638 | -1.472 | -1.861 | 0.277 | 1.112 |
| Q-whitening | **-0.588** | 0.403 | **-0.901** | **-1.067** | 0.854 | **-1.648** | **-1.574** | **-1.904** | 0.227 | **1.109** |

Table 10: Test RMSE and NLL on 10 univariate regression datasets (lower is better) using **SVGP-DKL**. We **compare learning inducing points in the feature space with learning inducing points in the input space**. Results are averaged over 10 random train/validation/test splits. Best ones with statistical significance are bold. See App. B.1.6 for more details.

| **Test RMSE** | Pol | Elevators | Bike | Kin40k | Protein | Keggdir | Slice | Keggundir | 3Droad | Song |
|---|---|---|---|---|---|---|---|---|---|---|
| feature space | 0.0867 | **0.347** | 0.067 | 0.086 | 0.614 | **0.0905** | 0.0237 | 0.122 | 0.382 | **0.772** |
| input space | **0.0614** | 0.343 | 0.013 | 0.067 | 0.598 | 0.0864 | 0.0189 | 0.120 | 0.329 | 0.773 |

| **Test NLL** | Pol | Elevators | Bike | Kin40k | Protein | Keggdir | Slice | Keggundir | 3Droad | Song |
|---|---|---|---|---|---|---|---|---|---|---|
| feature space | -1.014 | 0.361 | -1.292 | -1.070 | 0.931 | -1.007 | -2.214 | -0.691 | 0.464 | **1.161** |
| input space | **-1.388** | 0.347 | **-2.590** | -1.227 | 0.905 | -1.045 | -2.550 | -0.712 | 0.320 | 1.161 |

distribution. Here, we empirically compare the two whitening schemes and verify such claim. Table 8 summarizes a comparison of the **K**-whitening and **Q**-whitening in the DCSVGP model, and we observe that **Q-whitening outperforms K-whitening on 8 out of 10 datasets**. Table 9 shows similar results on the DCPPGPR model.

### B.1.6    DKL discussion

In the main paper, we evaluate SVGP-DKL, PPGPR-DKL and their decoupled counterparts SVGP-DCDKL and PPGPR-DCDKL on 10 UCI datasets, and we choose to learn inducing points in the input space for all models. Despite the convention of learning inducing points in the feature space, we found that for baseline models SVGP-DKL and PPGPR-DKL, learning inducing points in the input space performs better on the UCI datasets, see Table 10 and Table 11.

### B.2    Ablation Study on $\beta_2$

In addition to the ablation study results on DCPPGPR in the main paper, here we provide more ablation study results on DCSVGP and DCPPGPR with varying $\beta_2$ values on all datasets. Together with the representative results shown in the main paper, We obtain the same conclusion that

Table 11: Test RMSE and NLL on 10 univariate regression datasets (lower is better) using **PPGPR-DKL**. We **compare learning inducing points in the feature space with learning inducing points in the input space**. Results are averaged over 10 random train/validation/test splits. Best ones with statistical significance are bold. See App. B.1.6 for more details.

| Test RMSE | Pol | Elevators | Bike | Kin40k | Protein | Keggdir | Slice | Keggundir | 3Droad | Song |
|---|---|---|---|---|---|---|---|---|---|---|
| feature space | 0.1614 | **0.346** | 0.096 | 0.099 | 0.620 | **0.0953** | 0.0213 | **0.124** | 0.481 | **0.769** |
| input space | **0.0847** | 0.343 | **0.030** | **0.082** | **0.597** | 0.0871 | **0.0176** | 0.121 | **0.375** | 0.771 |

| Test NLL | Pol | Elevators | Bike | Kin40k | Protein | Keggdir | Slice | Keggundir | 3Droad | Song |
|---|---|---|---|---|---|---|---|---|---|---|
| feature space | -2.169 | **0.303** | -3.054 | **-1.754** | 0.764 | -1.574 | -2.118 | **-1.956** | 0.106 | **1.100** |
| input space | **-2.324** | 0.303 | **-3.096** | -1.739 | **0.712** | **-1.627** | **-2.625** | -1.951 | **-0.232** | 1.098 |

Table 12: Test RMSE and NLL on 10 regression datasets (lower is better). **We compare DCSVGP with different $\beta_2$ values**. Results are averaged over 10 random train/validation/test splits. Best ones with statistical significance are bold. We observe that $\beta_2 = 0.001$ or even $\beta_2 = 0$ are good choices. See App. B.2 for more details.

| Test RMSE | Pol | Elevators | Bike | Kin40k | Protein | Keggdir | Slice | Keggundir | 3Droad | Song |
|---|---|---|---|---|---|---|---|---|---|---|
| $\beta_2 = 1.0$ | 0.255 | **0.379** | 0.290 | 0.158 | 0.640 | **0.087** | **0.039** | **0.121** | 0.452 | 0.780 |
| $\beta_2 = 0.001$ | **0.156** | **0.379** | **0.286** | **0.150** | **0.604** | 0.086 | **0.039** | **0.121** | 0.434 | 0.777 |
| $\beta_2 = 0$ | **0.155** | **0.379** | **0.286** | 0.151 | 0.605 | **0.087** | **0.039** | **0.121** | 0.433 | 0.778 |

| Test NLL | Pol | Elevators | Bike | Kin40k | Protein | Keggdir | Slice | Keggundir | 3Droad | Song |
|---|---|---|---|---|---|---|---|---|---|---|
| $\beta_2 = 1.0$ | 0.120 | **0.450** | 0.184 | -0.421 | 0.974 | **-1.043** | -1.274 | **-0.695** | 0.632 | 1.170 |
| $\beta_2 = 0.001$ | **-0.373** | **0.450** | 0.165 | **-0.502** | 0.919 | **-1.047** | -1.293 | **-0.697** | 0.586 | 1.166 |
| $\beta_2 = 0$ | **-0.370** | **0.450** | 0.165 | -0.497 | 0.920 | -1.039 | -1.293 | **-0.698** | 0.582 | 1.167 |

1) $\beta_2 = 0.001$ or even $\beta_2 = 0$ are good choices for both DCSVGP and DCPPGPR (see Table 12 and Table 13);

2) on rare cases (only on the Pol and Elevators dataset, see Table 13), DCPPGPR could overfit and $\beta_2 = 0.1$ or $\beta_2 = 1.0$ would fix the issue and improve the prediction performance.

More detailed results and discussions follows.

**1D Example**  Using the same 1D toy example from the main paper, here in Figure 7 we show ablation study on this 1D example. Consistently, DCSVGP outperforms SVGP with 10 inducing points, and DCSVGP keeps improving as $\beta_2$ decreases. We also observe that the decoupled lengthscales $l_{\mathrm{mean}}$ and $l_{\mathrm{covar}}$ differ more and more as $\beta_2$ decreases and thus allowing the model to better fit mean and covariance differently.

**DCSVGP**  From Table 12, we observe that, DCSVGP favors small $\beta_2$ values and $\beta_2 = 0.001$ or even $\beta_2 = 0$ are good choices. The general trend is that DCSVGP performs better as $\beta_2$ decreases. To clearly see this trend, in Figure 8 we provide detailed performance study on 3 datasets: the 1D synthetic dataset, the Pol dataset, and the Song dataset. We observe that both RMSE and NLL decrease as $\beta_2$ decreases, and the learned lengthscales differ more and more as $\beta_2$ decreases to better model the mean and covariance differently.

**DCPPGPR**  From Table 13, we observe similar trend that DCPPGPR also favors small $\beta_2$ values and $\beta_2 = 0.001$ or $\beta_2 = 0$ are still good choices. In the main paper, we observe that DCPPGPR overfits on dataset Elevators and Protein, and on Protein we show how varying $\beta_2$ improves test NLL in detail (Figure 2 in the main paper). Here we also observe from Table 13 that regularization with $\beta_2 = 1.0$ would fix the issue for both dataset Elevators and Protein. Figure 13 shows how the test RMSE and NLL change with $\beta_2$ on the Elevators dataset – we observe that RMSE does not change much as $\beta_2$ varies, but NLL decreases as $\beta_2$ increases, showing how regularization improves test NLL.

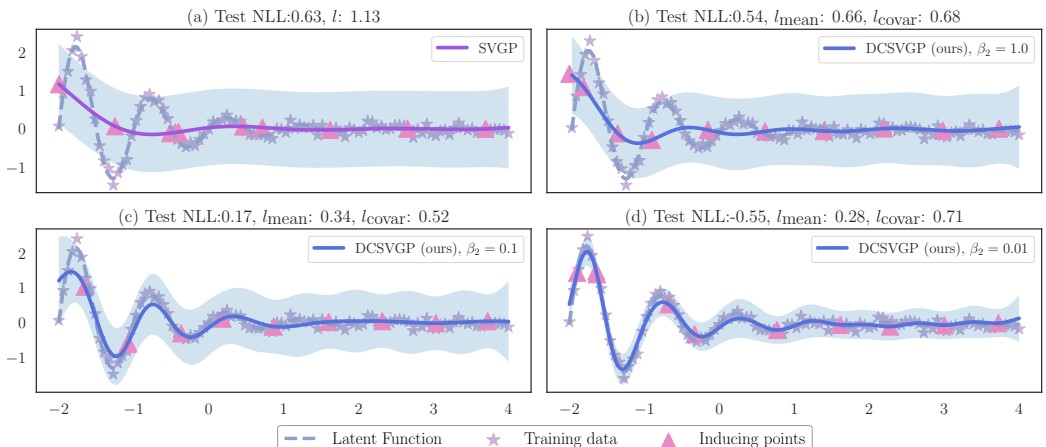

Figure 7: We compare model fit on a 1D latent function using 100 training samples and 10 inducing points. Solid curves with shading area depict the predictive mean and 95% confidence interval. In **subplot (a)**, SVGP underfits the latent function with large lengthscale $l = 1.13$. From **subplot (b) to (d)**, DCSVGP fits better and better with decreasing $\beta_2 = \{1.0, 0.1, 0.01\}$. We also observe that the decoupled lengthscales $l_{\text{mean}}$ and $l_{\text{covar}}$ differ more and more as $\beta_2$ decreases and thus allowing the model to better fit mean and covariance differently. See App. B.2 for more details.

Table 13: Test RMSE and NLL on 10 regression datasets (lower is better). **We compare DCPPGPR with different $\beta_2$ values.** Results are averaged over 10 random train/validation/test splits. Best ones with statistical significance are bold. We observe that DCPPGPR generally favors small $\beta_2$ values, and regularization would fix rare overfitting (on Elevators and Protein). See App. B.2 for more details.

| **Test RMSE** | Pol | Elevators | Bike | Kin40k | Protein | Keggdir | Slice | Keggundir | 3Droad | Song |
|---|---|---|---|---|---|---|---|---|---|---|
| $\beta_2 = 1.0$ | 0.258 | **0.393** | 0.366 | 0.233 | 0.645 | **0.090** | 0.044 | 0.124 | 0.546 | 0.778 |
| $\beta_2 = 0.001$ | **0.178** | 0.395 | 0.348 | 0.226 | 0.632 | 0.089 | 0.042 | 0.124 | 0.543 | 0.779 |
| $\beta_2 = 0$ | **0.177** | 0.393 | 0.345 | 0.227 | 0.634 | **0.090** | 0.043 | 0.123 | 0.544 | 0.779 |

| **Test NLL** | Pol | Elevators | Bike | Kin40k | Protein | Keggdir | Slice | Keggundir | 3Droad | Song |
|---|---|---|---|---|---|---|---|---|---|---|
| $\beta_2 = 1.0$ | -0.287 | **0.377** | -0.815 | -0.987 | **0.790** | -1.616 | -1.524 | -1.859 | 0.237 | **1.108** |
| $\beta_2 = 0.001$ | **-0.588** | 0.403 | **-0.901** | **-1.067** | 0.854 | -1.648 | -1.574 | -1.904 | 0.227 | 1.109 |
| $\beta_2 = 0$ | **-0.616** | 0.399 | -0.899 | -1.065 | 0.855 | -1.635 | -1.538 | -1.906 | 0.225 | 1.110 |

Table 14: Test RMSE on 10 regression datasets (lower is better). We **compare NNSVGP to DCSVGP and SVGP** and **compare NNPPGPR to DCPPGPR and PPGPR**. Results are averaged over 10 random train/validation/test splits. Best ones with statistical significance are bold. NNSVGP and NNPPGPR both yield the best test RMSE on 9 out of 10 datasets.

| | Pol | Elevators | Bike | Kin40k | Protein | Keggdir | Slice | Keggundir | 3Droad | Song |
|---|---|---|---|---|---|---|---|---|---|---|
| SVGP | 0.313 | 0.380 | 0.294 | 0.186 | 0.662 | 0.089 | 0.131 | 0.122 | 0.511 | 0.797 |
| DCSVGP | 0.156 | 0.379 | 0.286 | 0.150 | 0.604 | 0.086 | 0.039 | 0.121 | **0.434** | **0.777** |
| NNSVGP | **0.069** | **0.353** | **0.057** | **0.103** | **0.590** | **0.084** | **0.019** | **0.117** | 0.523 | 0.778 |
| PPGPR | 0.306 | 0.392 | 0.377 | 0.282 | 0.659 | 0.091 | 0.205 | 0.125 | **0.552** | 0.780 |
| DCPPGPR | 0.178 | 0.395 | 0.348 | 0.226 | 0.632 | 0.089 | 0.042 | 0.124 | **0.543** | 0.779 |
| NNPPGPR | **0.069** | **0.352** | **0.057** | **0.118** | **0.604** | **0.085** | **0.021** | **0.118** | 0.623 | **0.776** |

## B.3 Plausible Extension

**RMSE results of NNSVGP** We provide the RMSE results of NNSVGP and NNPPGPR here in Table 14 to support the observation that they yield the best test RMSE on 9 out of 10 datasets.

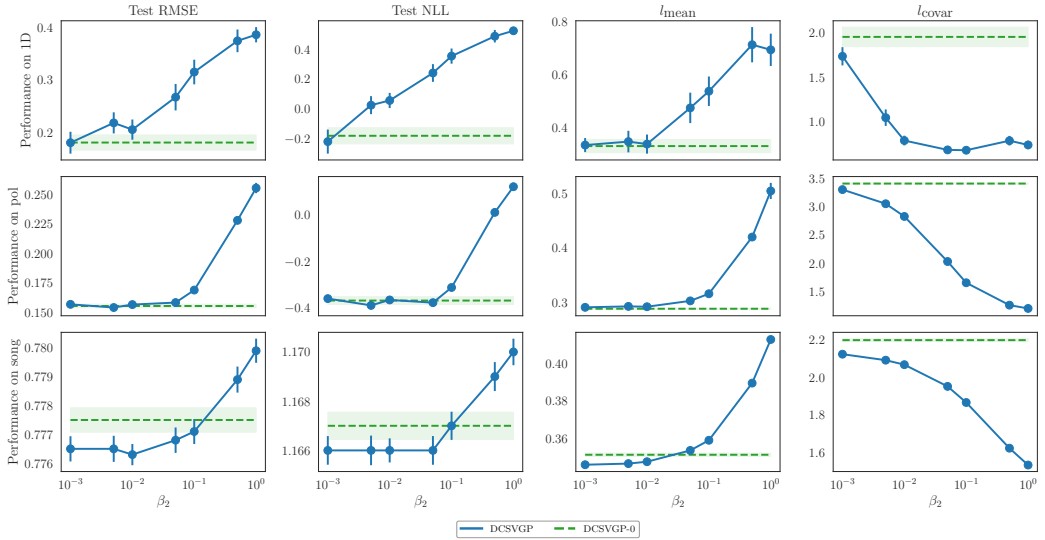

Figure 8: We evaluate **DCSVGP with varying** $\beta_2$ on the 1D dataset (first row), dataset Pol (second row) and Song (third row). Solid blue lines show DCSVGP with nonzero $\beta_2$ and green dashed lines show DCSVGP with $\beta_2 = 0$. First two columns contain test metrics, lower the better. Last two columns show the learned decoupled lengthscales $l_{\text{mean}}$ and $l_{\text{covar}}$. **We observe that both RMSE and NLL decrease as $\beta_2$ decreases**. See App. B.2 for more details.

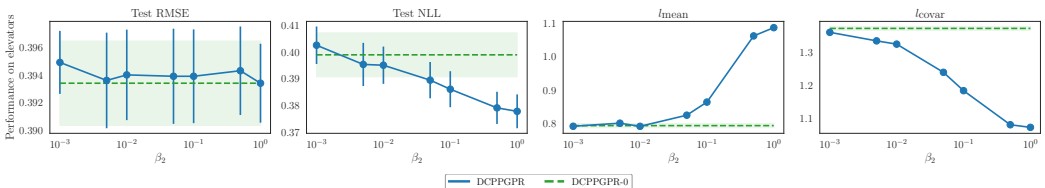

Figure 9: We evaluate **DCPPGPR with varying** $\beta_2$ on dataset Elevators, similar to Figure 8. We observe that test RMSE does not change much as $\beta_2$ varies, but NLL decreases as $\beta_2$ increases, showing how regularization improves test NLL. See App. B.2 for more details.

