# OpenReview forum: "Variational Gaussian Processes with Decoupled Conditionals"
_NeurIPS.cc/2023/Conference — NeurIPS 2023 poster_

### Official Review · Reviewer_TCkp · 2023-07-01

**Soundness:** 3 good
**Presentation:** 1 poor
**Contribution:** 2 fair
**Rating:** 3
**Confidence:** 3

**Summary:**

This paper addresses enhancing the performance of sparse approximate Gaussian processes, which has scalability with the dataset size but causes performance degradation.
The authors proposed a novel parametrization of conditionals as decoupled forms for training and testing ones, where lengthscales in kernel representations in their means and covariances can have different values.
Experimental results on multiple public datasets demonstrate that the proposed method performs better than existing methods.

**Strengths:**

- Experimental results of the proposed method showed promising performance.

**Weaknesses:**

### Originality:
- Originality is limited to the proposal of Eq.3.

### Clarity:
- The important concept of how to select or generate inducing points is not described in the main text. For example, inducing points are different between the left and right in Figure 1, but it is not described why.
- It is better to provide the definition of SVGP in Introduction.
- In the equation under l.53, some variables, such as mu, are not defined. The notation for x* is inconsistent
- In the description of Eq.1, it is better to define Kmm explicitly. Accordingly, the difference between Kmms for the proposed method and the existing methods is not clear.
- It is better to introduce p(f∗|fm) with more intuition.
- It is unclear how to determine the regularization parameter.

**Questions:**

NA

**Limitations:**

There is a study of performances over different hyperparameters.

---

> ### Author Rebuttal · Authors · 2023-08-08
>
> Thank you for your feedback. We will address each of your questions below.
>
> > Originality is limited to the proposal of Eq.3.
>
> We would greatly appreciate it if you can **concretely** point out the limitations on our novelty. We believe that referring to Eq (3) as our full novelty is reductive, as it describes only the modifications we make to the prior and not the resulting considerations when deriving an inference scheme or implementation details like whitening. Arguably, you could be this reductive about other prior published work just as easily, and in our opinion just as wrongly (e.g., by this logic, Salimbeni et al., 2019 “reduces” to equation 12, Jankowiak et al., 2020 “reduces” to equation 18, and so on).
>
> To be concrete about our novelty from our end, as far as we are aware, ours is not the first work to consider modifications  to the conditionals used in inducing point methods, but *is* the first to consider *more flexible* conditionals. This has significant payoff, reducing error by as much as half on several commonly used benchmark datasets in the GP literature. By being the first to do so, our method is arguably “novel.”
>
> > The important concept of how to select or generate inducing points is not described in the main text. For example, inducing points are different between the left and right in Figure 1, but it is not described why.
>
> In all results, the inducing points are learned automatically by maximizing the ELBO, which is (1) standard in practice, and (2) the default approach to selecting inducing points in software (e.g., both in GPFlow and in GPyTorch). This allows us to select the “best” inducing points, assuming the numerical optimization of the ELBO is done properly.
>
> It is natural and expected to have different inducing points in the left and right in Figure 1. The inducing points are learned parameters of the model, and the left and right figures depict different trained models. We will revise the paper to make this more clear.
>
> > It is better to provide the definition of SVGP in Introduction.
>
> SVGP, stochastic variational Gaussian process, is a widely used method in the literature and we will add the definition of SVGP in the introduction to make it more clear.
>
> > In the equation under l.53, some variables, such as mu, are not defined. The notation for x* is inconsistent.
>
> We will carefully go through the paper for better consistency. In the equation under l.53, $\mu(\cdot)$ is the prior mean function. We will write the variable $\mathbf{x}^*$ in l.53 in bold for consistency. Thanks for spotting the typo.
>
> > In the description of Eq.1, it is better to define Kmm explicitly. Accordingly, the difference between Kmms for the proposed method and the existing methods is not clear.
>
> $K_{mm} \in \mathbb{R}^{m\times m}$ is the kernel matrix formed by kernel function evaluated at inducing points: $K_{mm}(i, j) = k(u_i, u_j)$ where $u_i, u_j$ are inducing points, which is a standard expression in GP literature and we will make it more clear in the text.
>
> The difference between Kmms for our method and the existing methods lies in the decoupling of the kernel hyperparameters and we provide explicit examples in section 3.2 (example 1 and 2).
>
> > It is better to introduce $p(f^∗|f_m)$ with more intuition.
>
> $p(f^* \mid f_m)$ is the distribution of testing labels $f^*$ conditioned on the inducing values $f_m$ induced directly by having a GP prior. It would be the same distribution as conditioning a GP on the inducing points and the noise-free labels $f_m$ and making predictions at the test points.
>
> > It is unclear how to determine the regularization parameter.
>
> We thoroughly evaluate the effect of $\beta_2$ in Figure 2 and section 5.2 on two of our test datasets, with additional similar figures on most datasets in the supplementary materials (see l.207-l.209 in the paper and Figure 5&6 and Table 7&8 in the supplement). In practice, the lessons learned in this study seem to be general, where our method prefers lower values of $\beta_2$ but is insensitive otherwise. To set the regularization parameter even more precisely on a real world application of our method, consider doing cross validation.
>
> Please let us know if you have further questions and we would appreciate it if you would raise your score given our response.
>
> ### References
>
> Salimbeni, Hugh, et al. "Orthogonally decoupled variational Gaussian processes." Advances in neural information processing systems 31 (2018).
>
> Jankowiak, Martin, Geoff Pleiss, and Jacob Gardner. "Parametric gaussian process regressors." International Conference on Machine Learning. PMLR, 2020.

---

> ### Author Response · Authors · 2023-08-20
>
> Dear reviewer TCkp,
>
> As the discussion deadline is approching, we kindly ask you to check our rebuttal response :) If our rebuttal addressed your concerns, we would like to ask if are willing to reconsider your score. Also, please let us know if you have further questions we can response to! Thank you!

---

### Official Review · Reviewer_3gEr · 2023-07-04

**Soundness:** 4 excellent
**Presentation:** 3 good
**Contribution:** 3 good
**Rating:** 6
**Confidence:** 5

**Summary:**

This paper studies sparse Gaussian processes which allow different kernel hyperparameters such as length scale to be used, for instance, for the variational posterior mean and covariance. They derive a variational approximation that enables training of models of this kind. This extra scalable approximation flexibility is shown to lead to better predictions on UCI datasets, as well as a few benchmark-style Bayesian optimization problems.

EDIT: As indicated below, I've acknowledged the authors' rebuttal and previously updated my review and score appropriately.

**Strengths:**

The paper is technically sound, generally well-written, and includes all of the content and evaluation I would expect for this topic. The authors have done a reasonable and comprehensive job here. They evaluate on both modeling tasks (UCI dataset regression) and decision-making tasks (Bayesian optimization), as well as include an ablation study for one of the key non-obvious hyperparameters they introduce (the \beta regularization term which controls how the test conditional component enters the objective).

My most common reaction while reading this paper was "that makes sense" - which is actually very good, but also means that I don't have too many comments on what to add since the paper was largely clear, and therefore this review will be a bit short. The authors should not be discouraged by this - the shortness in this case should be understood as a sign they wrote the paper well.

**Weaknesses:**

This paper is studying what I view as a relatively old-school topic, and as a result much of the value here is from good evaluation and execution of ideas, rather than from the ideas themselves. Inducing point approximations have been studied very extensively, and at this stage there are so many variations out there that it isn't clear why we need to write more papers about them, since these methods are mature enough that readers can probably figure out new ones themselves when they are needed for applied work.

The need to add $\beta$ regularization parameters is a bit suspicious, since this suggests there might sometimes be a need to downscale the extra term coming from different conditionals because it starts behaving badly in the limit of many test points. It also forces the user to consider even more hyperparameters, and in most cases there are already enough of those out there. As a result, I'm curious for what choices of conditionals the respective infinite-dimensional variational approximation problem between stochastic processes is well-defined. See questions.

The empirical evaluation is overall good, but it focuses too much on RMSE tables on UCI datasets. These take up a lot of space and are not super informative, so it would be better to have just one and move extra results into the appendix, creating more room for describing the experiments. I would be particularly interested to see more figures which evaluate the technique's produced uncertainty, particularly since inducing points tend to blow up uncertainty and over-smooth the data in situations where there are not enough inducing points, and it would be much more interesting to see how the authors' technique behaves in this situation, rather than more RMSE tables.

**Questions:**

Is there a way to view the inducing point approximation you propose as a random function via pathwise conditioning? By this, I mean that, since the true posterior satisfies $(f|y)(.) = f(.) + K_{(.)x} (K_{xx} + \sigma^2 I)^{-1} (y - f(x) - \epsilon)$ where $f \sim GP(0,k)$ and $\epsilon \sim N(0,\sigma^2 I)$, and the approximation of Titsias satisfies $(f|u)(.) = f(.) + K_{(.)z} K_{zz}^{-1} K_{zx} (K_{xz} K_{zz}^{-1} K_{zx} + \sigma^2 I)^{-1} (y - g(x) - \epsilon)$ where $g(x) \sim N(0, K_{xz} K_{zz}^{-1} K_{zx})$ and $\epsilon \sim N(0,\sigma^2 I)$, your posterior approximation might admit a similar formula, and its structure should give insight into the approximation's behavior. Do you know what the respective formula is?

Is your variational optimization problem consistent in the limit of infinitely many test points? By this, I mean, does it lead to a valid Kullback-Leibler divergence minimization problem between stochastic processes, where the true posterior and variational approximation are absolutely continuous so that the KL is well-defined?

**Limitations:**

Yes

---

> ### Author Rebuttal · Authors · 2023-08-08
>
> Thank you for your positive feedback. We will address each of your questions below.
>
> > "Could move some RMSE tables into appendix and describe more about experiments."
>
> Thanks for the suggestion. We plan to use the additional page of content to add additional description for the experiments. We agree that the RMSE results are generally less important than the NLL results given the probabilistic nature of the model, and are happy to move them to create additional space as well.
>
> > " ... more figures which evaluate the technique's produced uncertainty, particularly since inducing points tend to blow up uncertainty and over-smooth the data in situations where there are not enough inducing points, and it would be much more interesting to see how the authors' technique behaves in this situation, rather than more RMSE tables."
>
> We agree that uncertainty evaluation is very important. We do at least provide the NLL metric that evaluates both accuracy and uncertainty. For visualization, with synthetic data, we see that when there are not enough inducing points, uncertainty blows up and over-smoothed data (Figure 1a) and our method resolves this issue (Figure 1b). On real datasets, generally our setting does not have enough inducing points due to large number of trainig data, and we provide model calibration results in terms of z-score distribution figures in the supplement (Figure 3 in section 2.1.3 in the supplement). It shows that our method has a better calibrated predictive uncertainty. We will draw more attention to this using the additional page. Please let us know if you would like to see more visualization of the uncertainty evaluation.
>
> > "View the inducing point approximation you propose as a random function via pathwise conditioning. ... Your posterior approximation might admit a similar formula, and its structure should give insight into the approximation's behavior. Do you know what the respective formula is?"
>
> Yes. We can derive a similar formula following Titsias. For standard variational GP, it requires computing the optimal variational distribution $\phi^*$ by differentiating the ELBO. Our ELBO for the decoupled model is formed similarly with an additional expected KL divergence term. If we set the regularization parameter $\beta_2=0$ (removing the additional expected KL divergence term), we could follow Titsias and find a similar optimal variational mean $ \mathbb{m}^*=\sigma^{-2} Q_{zz}\Sigma^{-1} Q_{zx} y$, where $\Sigma = Q_{zz} K_{zz}^{-1} Q_{zz} + \sigma^{-2} Q_{zx} Q_{xz}$. With $\mathbb{m}^*$, we can then derive the predictive mean $ \mu(\cdot) = Q_{(\cdot) z} Q_{zz}^{-1} \mathbb{m}^* $.
>
> Finally, the random function expression of the decoupled model is: $ (f\|y)(\cdot) = f(\cdot) + Q_{(\cdot) z} P_{zz}^{-1} Q_{zx} (Q_{xz} P_{zz}^{-1} Q_{zx} + \sigma^2 I)^{-1} (y - g(x) - \epsilon)$, where $P_{zz} = Q_{zz} K_{zz}^{-1} Q_{zz}$ and $g(x) \sim \mathcal{N}(0, Q_{xz} P_{zz}^{-1} Q_{zx})$.
>
> So the differences between the standard variational GP formula and ours are (1) our formula mostly involves the Q kernel matrices from decoupled mean and (2) the $K_{zz}$ matrix in the standard formula is replaced by $P_{zz}$ which involves the interaction of $K_{zz}$ and $Q_{zz}$.
>
> However, in the more general case where $\beta_2 \ne 0$, it is harder to solve for the optimal variational distribution and therefore trickier to find such a formula. With that said, we note (admittedly loosely) that as $\beta_2 \to \infty $, $K_{zz} \to Q_{zz}$ and therefore $P_{zz} \to K_{zz}=Q_{zz}$, which would recover the original derivation. Thus the closed form looks like–but may not actually be–a strict generalization.
>
>
> > "Is your variational optimization problem consistent in the limit of infinitely many test points? By this, I mean, does it lead to a valid KL divergence minimization problem between stochastic processes, where the true posterior and variational approximation are absolutely continuous so that the KL is well-defined?"
>
> Note that the modification we made to derive our model is not directly made to the SVGP ELBO, but to the GP prior itself. Thus, our derivation results in simply a standard variational inference problem for a slightly different probabilistic model. Our method is therefore best interpreted not as a “modified SVGP” but rather as a “modified GP” to which we are applying variational inference, and there is indeed still a true posterior being approximated by our KL minimization.
>
> Thank you for your insightful comments and please let us know if you have further questions.

---

> > ### Comment · Reviewer_3gEr · 2023-08-14
> >
> > Thanks very much for your comments! Overall, I liked the paper, I am somewhat surprised that my review turned out to be the most positive one. I have no major issues with the paper being accepted, so it seems to me that responding to the other referees' feedback and concerns is in order.

---

> > > ### Author Response · Authors · 2023-08-20
> > >
> > > Thank you for your support!

---

### Official Review · Reviewer_2Aqn · 2023-07-04

**Soundness:** 3 good
**Presentation:** 2 fair
**Contribution:** 3 good
**Rating:** 6
**Confidence:** 3

**Summary:**

This paper develops a variational approximation for learning sparse Gaussian Processes. The central idea is to "decouple" the mean and covariance parameters. From my read, this decoupling simply means formulating two different kernel matrices (called $Q$ and $K$ in the paper), and then working those new parameters through the inference and learning mechanisms: to that end, the paper provides appropriately-detailed derivations, including for some empirical concerns (such as whitening). The paper formulates the proposed approach in two cases, evaluating on 10 benchmark regression tasks. Overall, the proposed approach yields improved benchmark tasks. Ablation studies are also performed, which help provide insight into the approach.

**Strengths:**

Overall, this paper achieves an acceptable balance of mathematical derivation vs. explanation. However, please see "Clarity" in weakness for qualifications.

Experiments: Experiments were averaged across 10 runs, and sufficient details of standard error & significance were included. There's an appropriate amount of rigor, and ablation studies run.

**Weaknesses:**

Clarity: Key summary statements, to act as milestones, throughout would definitely help. This is especially true of Table 1, section 3.2 (Eqn 3), and section 3.3: broadly, the equations are presented, but without a lot of discussion contextualizing the meaning and importance of those equations.

Limited discussion: Given the number of experiments run, the discussion does not have a good pay-off. For example, Table 5 (one of the ablation studies) has fewer than 3 lines of prose dedicated to discussing those results in the main paper.

Lack of error analysis: The battery of regression benchmark tasks gives some evidence of broad performance, but there's no error analysis or insight into the numbers and performance. Currently, the results read as "just numbers": it's not clear, e.g., what a reduction from 0.321 to 0.156 on "Pol" means.

This is more of a minor point, but not without impact: splitting related work into sections 2.1 and 4 was not effective for me. It made 2.1 very much a slog, and section 4 seem unimportant.

**Questions:**

Q1: Please provide a few paragraphs that would form the basis of an error analysis & extended discussion section.

Q2: Please provide a limitation discussion (or if I missed it in the paper, please point me to it).

**Limitations:**

No: no obvious limitations section was provided in the main paper.

---

> ### Author Rebuttal · Authors · 2023-08-08
>
> Thank you for your positive feedback. We reply to each of your suggestions and questions below.
>
> > Clarity: Key summary statements, to act as milestones, throughout would definitely help. This is especially true of Table 1, section 3.2 (Eqn 3), and section 3.3: broadly, the equations are presented, but without a lot of discussion contextualizing the meaning and importance of those equations.
>
> Thank you for pointing out areas we could focus on for adding additional clarification. We found space to be fairly tight to include our reasonably extensive experimental evaluation, and plan to use the additional content page to add additional summary and discussions.
>
> > Limited discussion: Given the number of experiments run, the discussion does not have a good pay-off. For example, Table 5 (one of the ablation studies) has fewer than 3 lines of prose dedicated to discussing those results in the main paper.
>
> We will make sure to add more discussion of experimental results, especially for Table 4 and 5. For example, in Table 5, our SVGP-DCDKL performs the best in both mean performance (RMSE) and probabilistic performance (NLL). This suggests that it is more beneficial to have different models (decoupled feature extractors in SVGP-DCDKL) for conditional mean and covariance rather than only modeling the conditional mean (SVGP-MeanDKL) or having same models (SVGP-DKL). Moreover, it is surprising that SVGP-MeanDKL outperforms baseline SVGP-DKL, which re-emphasizes that a *decoupled* (*different*) or even a much simpler model for the covariance is more beneficial than using the same model for conditional mean and covariance.  For the PPGPR base model, PPGPR-DCDKL also performs the best and we can draw the same conclusion. However, PPGPR-MeanDKL does not outperform the baseline PPGPR-DKL. This suggests that for the PPGPR base model, it is vital to decouple the feature extractor (PPGPR-DCDKL) rather than having a simpler model for the covariance (PPGPR-MeanDKL) since PPGPR treats the predictive variance differently in the learning objective.
>
> > Lack of error analysis: The battery of regression benchmark tasks gives some evidence of broad performance, but there's no error analysis or insight into the numbers and performance. Currently, the results read as "just numbers": it's not clear, e.g., what a reduction from 0.321 to 0.156 on "Pol" means.
>
> We use standard regression benchmark datasets that are commonly used in GP regression papers (e.g. Wenger et al., 2022, Jankowiak et al., 2020, Wang et al., 2019), and the labels are standardized to mean 0 and variance 1. We will do our best to better explain the meaning of improvement in numbers. For example, the reduction from 0.321 to 0.156 on "Pol" generally indicates a large performance improvement.
>
> > This is more of a minor point, but not without impact: splitting related work into sections 2.1 and 4 was not effective for me. It made 2.1 very much a slog, and section 4 seem unimportant.
>
> We will do our best to reorganize these related work. Section 2.1 is intended to contain necessary background and prior work needed to understand our method directly. Section 4 is intended to contain broader related work and provide context. Such a section is commonly included and we will do our best to make it more coherent.
>
> > Please provide a few paragraphs that would form the basis of an error analysis & extended discussion section.
> Please provide a limitation discussion (or if I missed it in the paper, please point me to it).
>
> As stated above, we will provide more summary statements, more discussion of experiment results, as well as error analysis. We will also add a limitation discussion -- for example, we only applied our decoupling framework to regression tasks and BO applications, and we have not applied it to classification tasks.
>
> Thank you for your positive and constructive feedbacks! Please let us know if you have further questions.
>
>
> ### References
>
> Wenger, Jonathan, et al. "Preconditioning for scalable Gaussian process hyperparameter optimization." International Conference on Machine Learning. PMLR, 2022.
>
> Jankowiak, Martin, Geoff Pleiss, and Jacob Gardner. "Parametric gaussian process regressors." International Conference on Machine Learning. PMLR, 2020.
>
> Wang, Ke, et al. "Exact Gaussian processes on a million data points." Advances in neural information processing systems 32 (2019).

---

> > ### Comment · Reviewer_2Aqn · 2023-08-21
> >
> > Thank you for your detailed response. Assuming these changes are included in the next version, they satisfactorily address my main concerns.

---

> ### Author Response · Authors · 2023-08-20
>
> Dear reviewer 2Aqn,
>
> As the discussion deadline is approching, we kindly ask you to check our rebuttal response :) Also, please let us know if you have further questions we can response to! Thank you!

---

### Official Review · Reviewer_u1uu · 2023-07-05

**Soundness:** 3 good
**Presentation:** 3 good
**Contribution:** 2 fair
**Rating:** 6
**Confidence:** 3

**Summary:**

This paper considers the problem of increasing the expressivity of sparse Gaussian processes without increasing the number of inducing points by considering a decoupled ELBO. In their setup, the gram matrices appearing in the predictive mean and covariance have different parameterizations. They consider two such parameterizations, the first corresponding to learning different length-scales in a non-ARD RBF kernel, and the second using different neural networks in a deep kernel learning setup. An ELBO is derived in this setup, and a version is given for the PPGPR analogue. Whitening is considered with mean-whitening being favored. Numerical examples are given on UCI regression and Bayesian optimization tasks. The effects of differing levels of regularization are examined. Extensions parameterized with further neural networks are considered.

**Strengths:**

The paper is well structured and written. Numerical examples are given and their methods appear to perform well. The new parameter that's introduced in their derivation is studied.

**Weaknesses:**

There are typos/grammatical errors, as well as inconsistencies in use of abbreviations (see e.g. 5.2 vs the previous paragraph), so this should be thoroughly checked by the authors. The non-DKL baselines would be illuminating in the DKL experiments. Only non-ARD kernels are considered when considering different length-scales without justification - it would seem natural to consider different length scales in each dimension as is common practice. The authors give choice between one of two different whitening matrices, however no justification is given to why both are not used. Since the model separates the posterior mean and covariance, a metric that measures both and their tradeoff would be useful, or at least discussion on how this can be inferred from the reported results.

**Questions:**

How do the DKL models compare to SVGP?
Why are only non-ARD kernels considered?
Why is a model that utilizes both whitening matrices not considered?
Can you report performance in a way that considers the tradeoff between accuracy and uncertainty?

**Limitations:**

These are discussed above from my end. No limitations of the study are listed in the work, however the risk of harm or civilisation collapse based on this work is minimal.

---

> ### Author Rebuttal · Authors · 2023-08-08
>
> Thank you for your feedback. We will address each of your questions below.
>
> > Typos/grammatical errors and inconsistencies in use of abbreviations.
>
> Thank you for pointing this out. We will make sure to resolve all inconsistencies, for example changing DKL-DCPPGPR to PPGPR-DCDKL in 5.2.
>
> > The non-DKL baselines would be illuminating in the DKL experiments. How do the DKL models compare to SVGP?
>
> The difference between non-DKL and DKL is only in the use of a feature extractor. If you wish to compare DKL to non-DKL performance, note that we run all methods on the same datasets using the same setup, so the values in Table 2 and 3 are already comparable.
>
> > Only non-ARD kernels are considered when considering different length-scales without justification.
>
> We do consider ARD kernels, and decouple the multi-dimensional length scales in the same way. We mentioned that results with ARD kernels are in the supplement (see l.179-l.181 in the paper and Table 2 in the supplement). Decoupled ARD kernels performs similarly well as decoupled non-ARD kernels. We will draw more attention to this.
>
> > The authors give choice between one of two different whitening matrices, however no justification is given to why both are not used.
>
> We can only choose to use one of the whitening schemes (see l.81-l.84 and l.146-l.150 in the paper). If one uses both whitening matrices, for example whitening the variational mean $\mathbb{m}$ with $K_{mm}$ and whitening the variational covariance $\mathbb{S}$ with $Q_{mm}$, then neither the predictive distribution nor the KL divergence will be simplified.
>
> We choose to use Q matrices and provide the intuition of our choice. We also provide empirical results to compare the two choices in the supplement (see l.179-l.181 in the paper, and Table 3&4 in section 2.1.5 in the supplement). We will draw more attention to this.
>
> > Since the model separates the posterior mean and covariance, a metric that measures both and their tradeoff would be useful, or at least discussion on how this can be inferred from the reported results.
>
> We did not separate the posterior mean and covariance directly. We provided a decoupled conditional probability, which still leads to a unified ELBO for a single probabilistic model. We already report both RMSE and NLL, with RMSE measuring only mean performance and NLL measuring probabilistic performance. Reporting these two metrics to address this concern is common practice in the GP literature (e.g. Hensman et al., 2017, Havasi et al., 2018, Salimbeni et al., 2019, Jankowiak et al., 2020).
>
> > Can you report performance in a way that considers the tradeoff between accuracy and uncertainty?
>
> A GP provides a natural tradeoff between accuracy and uncertainty since predictive mean and covariance are analytically computed. We evaluate two metrics - RMSE measuring the predictive mean (accuracy) and NLL measuring both the predictive mean (accuracy) and predictive variance (uncertainty).
>
> If our response answers you questions, would you kindly update your rating? Please let us know if you have further questions.
>
>
>
>
> ### References
>
> Hensman, James, Nicolas Durrande, and Arno Solin. "Variational Fourier Features for Gaussian Processes." J. Mach. Learn. Res. 18.1 (2017): 5537-5588.
>
> Havasi, Marton, José Miguel Hernández-Lobato, and Juan José Murillo-Fuentes. "Deep Gaussian processes with decoupled inducing inputs." arXiv preprint arXiv:1801.02939 (2018).
>
> Salimbeni, Hugh, et al. "Orthogonally decoupled variational Gaussian processes." Advances in neural information processing systems 31 (2018).
>
> Jankowiak, Martin, Geoff Pleiss, and Jacob Gardner. "Parametric gaussian process regressors." International Conference on Machine Learning. PMLR, 2020.

---

> > ### Comment · Reviewer_u1uu · 2023-08-20
> > **Response**
> >
> > I've read your response and am satisfied with the document provided the promised changes are made. While introducing a new metric to directly measure tradeoff between uncertainty and accuracy would be nice, doing so is probably out of the context of this work and reporting RMSE and NLL is reasonable. I have adjusted my score accordingly.

---

> ### Author Response · Authors · 2023-08-20
>
> Dear reviewer u1uu,
>
> As the discussion deadline is approching, we kindly ask you to check our rebuttal response :) If our rebuttal addressed your concerns, we would like to ask if are willing to reconsider your score. Also, please let us know if you have further questions we can response to! Thank you!

---

### Decision · Program_Chairs · 2023-09-21

**Decision:**

Accept (poster)

**Comment:**

The paper proposes a variational approximation for sparse GP by using the decoupled kernel parameters for posterior mean and covariance.  Improved performance is demonstrated on 10 benchmark regression tasks with ablation and on a Bayesian optimization task.

Reviewers acknowledged the technically sound contributions that bring good performance compared to baselines, while they raised major concerns:

- Only non-ARD kernels are considered as baselines. (actually some in supplementary)
- No justification for some design choices
- Clarity issues
- lack of error analysis
- Limited originality

The authors addressed all major concerns, and all reviewers (except an unengaged one) agree to accept the paper.